# Modulation of aperiodic EEG activity provides sensitive index of cognitive state changes during working memory task

Tisa Frelih[1,2,3†], Andraž Matkovič[3,4*†], Tjaša Mlinarič[3,5], Jurij Bon[1,2,3], Grega Repovš[4]

[1]University Psychiatric Clinic Ljubljana, Slovenia, Ljubljana, Slovenia; [2]Faculty of Medicine, University of Ljubljana, Ljubljana, Slovenia; [3]Department of Neurology, University Medical Centre Ljubljana, Ljubljana, Slovenia; [4]Department of Psychology, Faculty of Arts, University of Ljubljana, Leuven, Belgium; [5]Department of Neuroscience, Laboratory for Neuro- and Psychophysiology, KU Leuven, Leuven, Belgium

*For correspondence:
andraz.matkovic@ff.uni-lj.si

†These authors contributed equally to this work

## eLife Assessment

This **valuable** work explores the timely idea that aperiodic activity in human electrophysiology recordings is dynamically modulated in response to task events in a manner that may be relevant for behavioral performance. Moreover, the authors present **solid** evidence that, in some circumstances, these aperiodic changes might be misinterpreted as oscillatory changes.

**Abstract** To better understand cognitive processes, it is valuable to consider both periodic (oscillatory) and aperiodic electrophysiological brain activity. The aim of this study was to clarify how the periodic and aperiodic electrophysiological components reflect the cognitive processes involved in working memory. Fifty-seven human participants performed an n-back task while their brain activity was recorded using EEG. The analysis of both components of the EEG spectrum during task performance indicates that both periodic and aperiodic activities exhibit distinct task-related spatiotemporal dynamics that are closely related to cognitive demands. The results suggest that a substantial portion of the changes often attributed to theta oscillations in working memory tasks may be influenced by shifts in the spectral slope of aperiodic activity. This finding indicates that the modulation of aperiodic activity, associated with cognitive control processes, could provide a more sensitive index of cognitive state changes than previously recognised. To further confirm our findings, we also used these analysis methods in an item-recognition task, which showed similar patterns of periodic and aperiodic activity. These observations challenge the conventional understanding of low-frequency oscillations in cognitive processing and raise concerns about the routine practice of EEG baseline correction in time-frequency analysis, as it may obscure significant modulations in continuous working memory tasks. Consequently, the inclusion of aperiodic activity as a fundamental component of EEG analysis is likely to be critical for the accurate representation of the neural basis of cognition.

## Introduction

Working memory is a crucial cognitive ability that enables goal-directed behaviour and higher cognitive function. It involves the maintenance and manipulation of information in the absence of sensory stimuli (*Baddeley, 2012*) and depends on a complex set of cognitive processes and neural mechanisms (*Wager and Smith, 2003*). Due to its central role in cognition, working memory has been extensively studied using electroencephalography (EEG). This method allows for time-frequency analysis of brain activity (*Cohen, 2014*), often focusing on predefined frequency bands such as theta (4–8 Hz), alpha (6–12 Hz), and beta (13–30 Hz). One widely used paradigm for investigating working memory via time-frequency analysis is the n-back task (*Owen et al., 2005*; *Gevins et al., 1997*), which integrates key working memory processes such as encoding, maintenance, updating, and recall (*Pesonen et al., 2007*; *Chen and Huang, 2015*).

Beyond band-specific spectral modulations, EEG power spectra exhibit a broadband decrease in power with increasing frequency, following a $1/f$ scaling pattern (where $f$ is frequency). This pattern is widely thought to reflect non-oscillatory (aperiodic) neural activity that lacks distinct spectral peaks. Because aperiodic activity spans a continuous frequency range, it can obscure changes in periodic (oscillatory) activity, complicating interpretation, unless appropriately separated or accounted for (*Cohen, 2014*; *Donoghue et al., 2020b*).

Given that aperiodic activity is sometimes regarded as background noise, a common approach in task-based EEG studies is to apply baseline correction to emphasize task-related changes in power. Baseline correction aims to normalise the data by removing this background activity, often using decibel or z-score normalisation. These methods typically assume stable aperiodic activity across time (*Gyurkovics et al., 2021*; *Merkin et al., 2023*; *Donoghue et al., 2020b*); however, in continuous paradigms like the n-back task, where no neutral pre-stimulus interval exists, this assumption may not hold. Moreover, recent studies show that ignoring task-related aperiodic fluctuations can distort estimates of periodic activity and mask meaningful physiological and behavioural information (*Gyurkovics et al., 2021*; *Donoghue et al., 2020a*; *Gyurkovics et al., 2022*; *Virtue-Griffiths et al., 2022*; *Kałamała et al., 2024*; *Akbarian et al., 2023a*).

Computational models suggest that these aperiodic fluctuations reflect the balance between excitation and inhibition (E:I) in the brain (*Gao et al., 2017*; *Lim and Goldman, 2013*), essential for efficient cognitive processing (*Ahmad et al., 2022*). Aperiodic activity is often approximated as linear in log-log space (*Donoghue et al., 2020b*; *Barry and De Blasio, 2021*). In the commonly used $1/f$ formulation, the aperiodic exponent describes how quickly power decreases as frequency increases. When both axes are shown on a logarithmic scale, the line's slope is the negative of this exponent, so a larger exponent yields a more steeply descending slope. While sometimes referred to as the 'aperiodic exponent,' we use the term 'slope' here, except where methodological descriptions require exponent notation. A steeper aperiodic slope (larger exponent; more negative slope) implies higher low-frequency power and lower high-frequency power, potentially indicating stronger cognitive engagement (*Gao et al., 2017*; *Akbarian et al., 2023b*), whereas a flatter slope (less negative exponent) can reflect more uniform power across frequencies (*Gyurkovics et al., 2022*; *Kałamała et al., 2024*). However, some tasks show flatter slopes even under increased engagement (*Waschke et al., 2021*; *Cunningham et al., 2023*). Moreover, task-related changes in spectral power within a defined frequency range, typically interpreted as oscillatory dynamics, may be significantly co-modulated by the aperiodic slope shift (*Akbarian et al., 2023a*; *Gyurkovics et al., 2022*; *Virtue-Griffiths et al., 2022*).

Given these findings, it is methodologically imperative to carefully parameterise both periodic and aperiodic components of an EEG before interpreting time-frequency analysis results. Common approaches to spectral or time-frequency analysis often assume that distinct peaks in the power spectrum reflect true oscillatory activity once the 1/f background is removed (*Donoghue et al., 2020a*; *Donoghue et al., 2020b*). However, the presence of power in a particular frequency band is not necessarily indicative of oscillatory dynamics, as aperiodic activity can constitute a significant proportion of observed power spectra (*Bullock et al., 2003*; *Donoghue et al., 2020a*; *Donoghue et al., 2022*). Moreover, time-frequency representations (TFRs) typically highlight temporal fluctuations in power (*Cohen, 2014*), but do not inherently differentiate between periodic and aperiodic contributions. If part of the TFR signal arises from aperiodic shifts, baseline correction may confound rather than clarify the nature of observed effects.

In this study, we examined how aperiodic and periodic EEG components each contribute to working memory demands in the n-back task, addressing potential biases introduced by baseline correction. Furthermore, we focused on disentangling how aperiodic slope is influenced by memory load, stimulus type, or task modality. We based our analyses on a dataset of middle-aged to older adults, half of whom reported subjective cognitive complaints; nonetheless, behavioural outcomes and electrophysiological measures did not differ between groups. To assess the generalisability of our findings, we applied the same analysis to two additional datasets which allowed us to test whether similar aperiodic–periodic dynamics emerge across different working memory contexts.

Our study builds on previous research by using high temporal resolution data to investigate the aperiodic slope across two modalities (visuospatial/verbal). In contrast to studies that use discrete time windows (*Akbarian et al., 2023a*; *Kałamała et al., 2024*), we analyse the activity at each time point, capturing short-lived slope fluctuations that could reveal crucial insights into continuous working memory processes. Although a similar approach has been used to track temporal dynamics in sleep and resting state (e.g. *Wilson et al., 2022*; *Ameen et al., 2024*), as well as in task-based contexts (e.g. *Barrie et al., 1996*; *Preston et al., 2025*), its specific application to working memory paradigms remains underexplored. While some evidence suggests modality-specific aperiodic variations (*Waschke et al., 2021*), we propose that the n-back task primarily recruits domain-general processes (*Owen et al., 2005*; *Gazzaley and Nobre, 2012*; *Olesen et al., 2004*), leading to similar slope modulations across both verbal and visuospatial modalities (*Gyurkovics et al., 2022*). We, therefore, predict that the aperiodic slope would be modulated by the processing demands of the n-back task, varying with load and stimulus type but not between visuospatial and verbal conditions.

By explicitly separating aperiodic and periodic components and tracking their dynamics over time, our study offers a novel perspective on the relationship between baseline correction and aperiodic activity in continuous tasks.

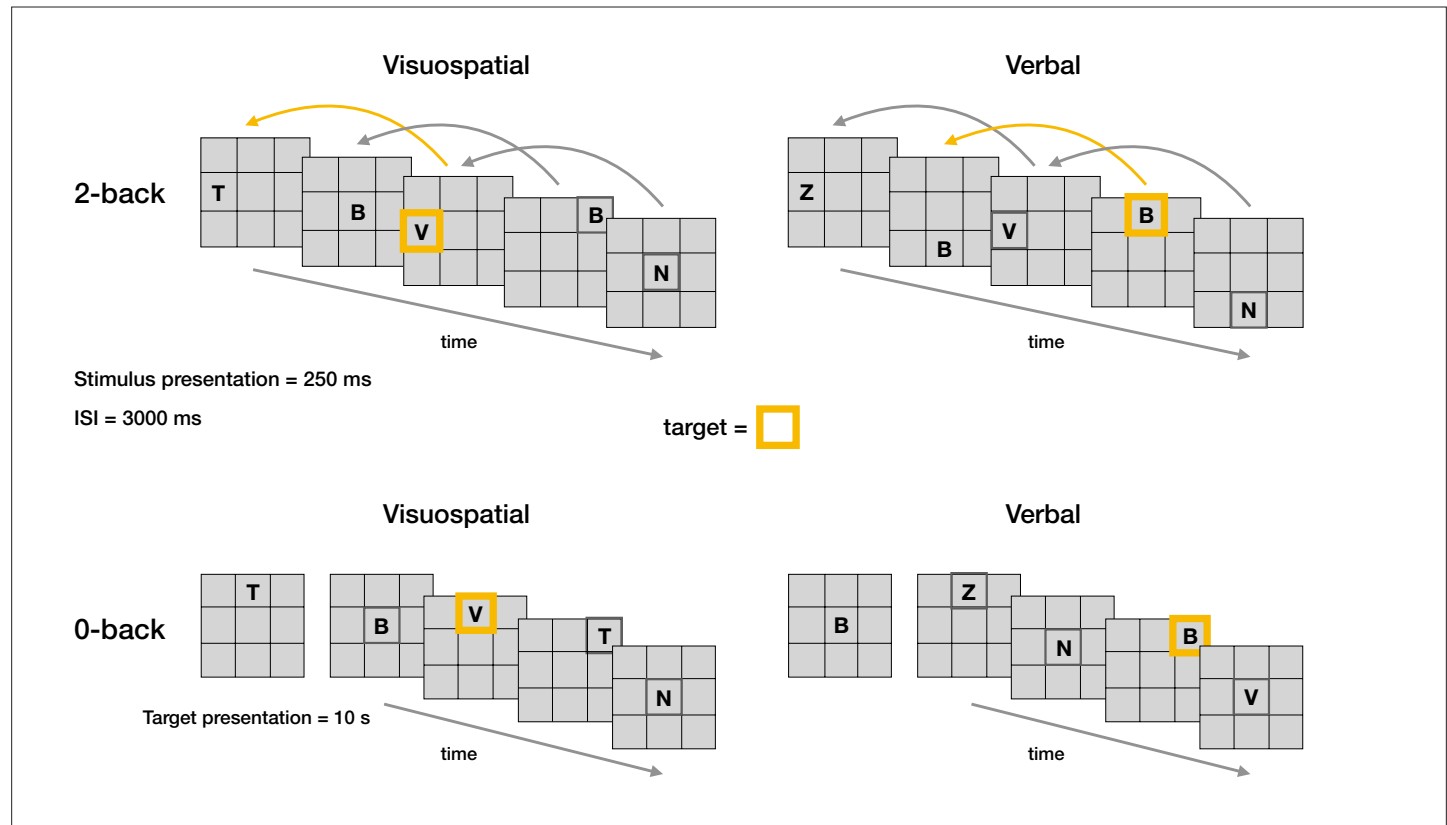

**Figure 1.** Schematic representation of the n-back task. The n-back working memory task was conducted in two distinct modalities: visuospatial and verbal. In the 2-back condition, participants had to identify whether the current stimulus matched the one presented two steps previously. In the visuospatial modality, the target was a spatial location, whereas in the verbal modality, the target was a letter. In the 0-back condition, participants' task was to respond to a predefined target, with the type of target corresponding to the task modality. Target stimuli are highlighted in orange.

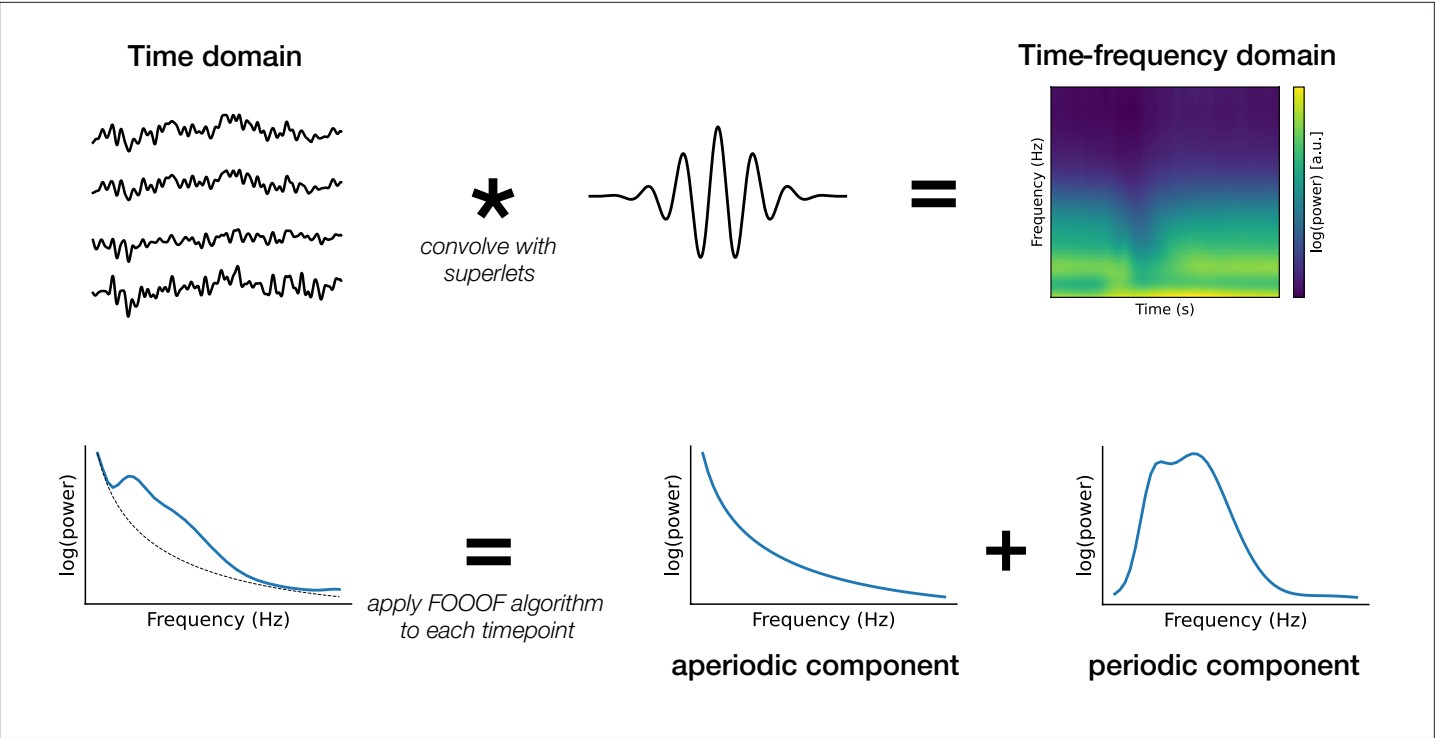

**Figure 2.** Schematic of the analysis. The time domain data were first transformed into the time-frequency domain by convolution with superlets (*Moca et al., 2021*). Next, the periodic and aperiodic components of the power spectrum density were estimated for each time point using FOOOF (Fitting Oscillations and One-Over-F) algorithm (*Donoghue et al., 2020b*). The aperiodic component was characterised by the aperiodic slope (the negative counterpart of the exponent parameter) and the offset, which together describe the underlying broadband spectral shape.

## Results

We analysed the EEG data from the visuospatial and verbal n-back tasks (*Figure 1*). We transformed the data into the time-frequency domain by convolving it with superlets (*Moca et al., 2021*). Subsequently, we applied the FOOOF (Fitting Oscillations and One-Over-F) algorithm (*Donoghue et al., 2020b*) to decompose the activity into periodic and aperiodic components (*Figure 2*). We estimated two aperiodic parameters using FOOOF: the offset and the exponent. While the FOOOF framework refers to this parameter as the exponent, we use the term 'aperiodic slope' throughout the manuscript for consistency with broader literature on spectral power scaling. Note that the exponent is numerically positive, whereas the slope is its negative counterpart (i.e. a steeper (more negative) slope in log-log space corresponds to a higher (positive) exponent value).

### Changes in aperiodic activity appear as low-frequency power in baseline-corrected time-frequency plots

As the first step of the analysis, we compared periodic and aperiodic activity with baseline-corrected time-frequency activity. In the baseline-corrected time-frequency plot, we observed a decrease in alpha and beta power up to 1 s post-stimulus, while low-frequency power increased from 0 s post-stimulus (*Figure 3*, *Figure 3—figure supplement 3*). The observed pattern was minimally affected by the choice of baseline (*Figure 3—figure supplement 2*). Specifically, whether we employed subtractive or divisive baseline normalisation (*Gyurkovics et al., 2021*), the qualitative pattern of alpha, beta, and low-frequency (delta and theta) changes remained consistent.

Note that although no prominent spectral peak in the theta range was observed at the group level, and some of this activity could potentially fall within the delta range, similar low-frequency patterns have often been referred to as 'theta' in previous work, even in the absence of a clear spectral peak (e.g. *Palomäki et al., 2012*; *Pesonen et al., 2007*; *Rossi et al., 2023*). To acknowledge both perspectives, we use the more neutral term 'low-frequency activity' throughout the paper, while noting its potential overlap with activity commonly described as frontal theta.

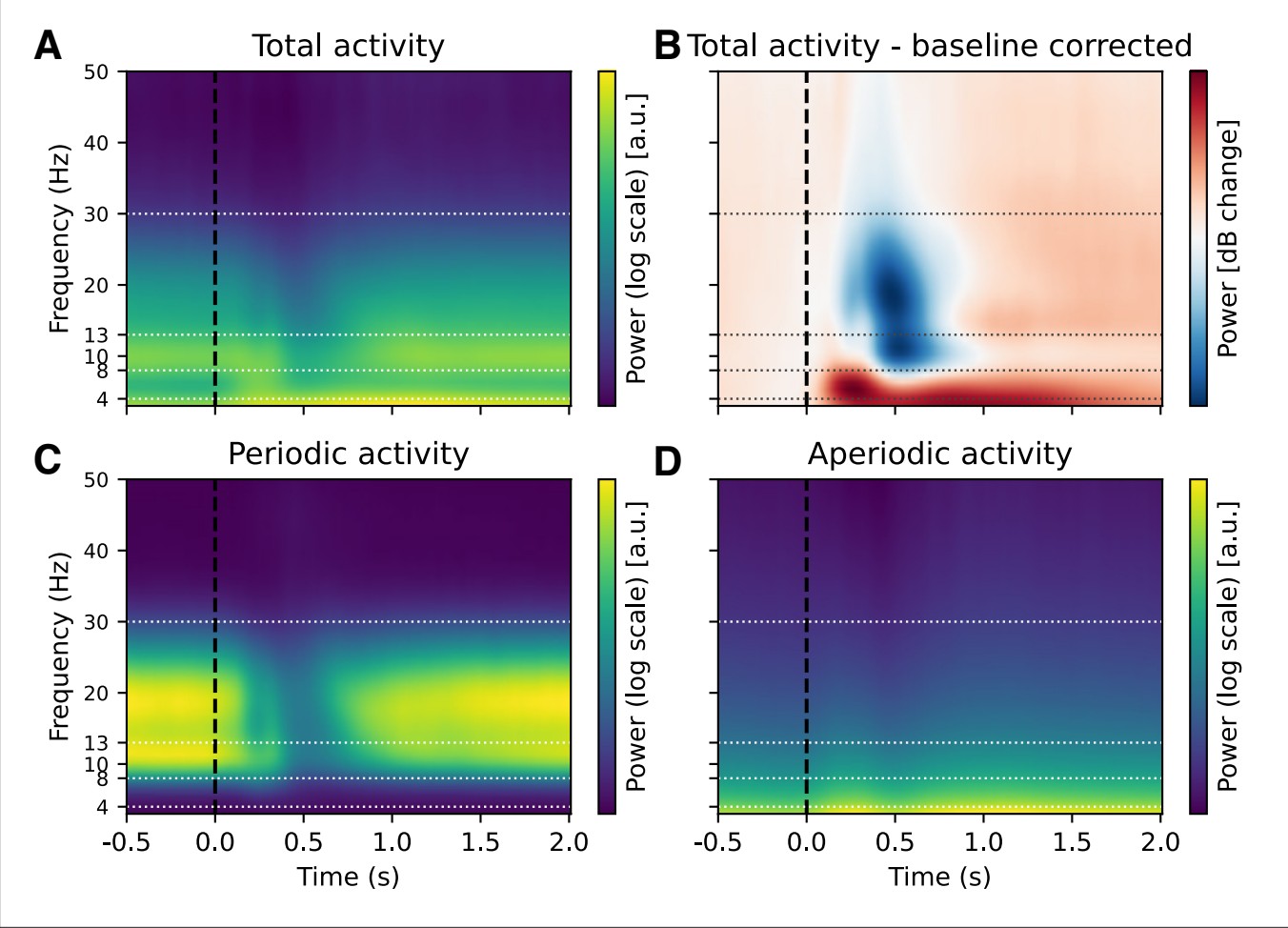

**Figure 3.** Effect of baseline correction and FOOOF decomposition on time-frequency decomposition in the n-back task. Illustration of the effect of baseline correction and FOOOF decomposition on electroencephalography (EEG) time-frequency analysis. (**A**) 'Raw' total power from time-frequency decomposition is difficult to interpret due to the $1/f$ power scaling of EEG power spectra (here representing average data across subjects, channels, and conditions). In panel B, we applied baseline correction (decibel conversion) using a pre-stimulus interval of –0.5 to –0.2 ss for comparison. The baseline correction showed a significant decrease in alpha and beta power from 0 to 1 s and a concomitant increase in low-frequency power lasting up to 2 ss. The observed changes were very similar across different choices of baseline correction (*Figure 3—figure supplement 2*). However, it's unclear from the baseline-corrected data whether the observed changes in the low-frequency range reflect periodic or aperiodic contributions. To disentangle these components, we decomposed the time-frequency signal into periodic and aperiodic contributions using spectral parameterisation. (**C**) The periodic component includes only the parameterised spectral peaks, reconstructed from Gaussian fits to the power spectrum (see Methods for details). (**D**) The aperiodic component reflects the 1/f-like background activity. This decomposition suggests that changes in power in the low-frequency (delta and theta) range may largely reflect changes in aperiodic activity. See *Figure 3—figure supplement 1* for the corresponding figure with a logarithmic y-axis.

The online version of this article includes the following figure supplement(s) for figure 3:

**Figure supplement 1.** Effect of baseline correction and FOOOF decomposition on time-frequency decomposition in the n-back task.

**Figure supplement 2.** The comparison of different baseline corrections on time-frequency decomposition results in the n-back task.

**Figure supplement 3.** Effect of baseline correction and FOOOF decomposition on time-frequency decomposition (control dataset).

**Figure supplement 4.** Effect of baseline correction and FOOOF decomposition on time-frequency decomposition (control dataset).

Notably, when we examined periodic activity, we only observed changes in alpha and beta frequency bands, while increases in the low-frequency range were absent (Figure 6). This suggests that low-frequency changes may, at least in part, reflect aperiodic activity rather than periodic (narrow-band) processes. To quantitatively assess this observation, we directly compared the FOOOF and baseline-corrected time-frequency decompositions by performing correlations for each channel-frequency-time point (*Figure 4*, *Figure 4—figure supplement 1*). We observed strong positive correlations between periodic power and baseline-corrected power in the alpha and beta ranges across all channels, but

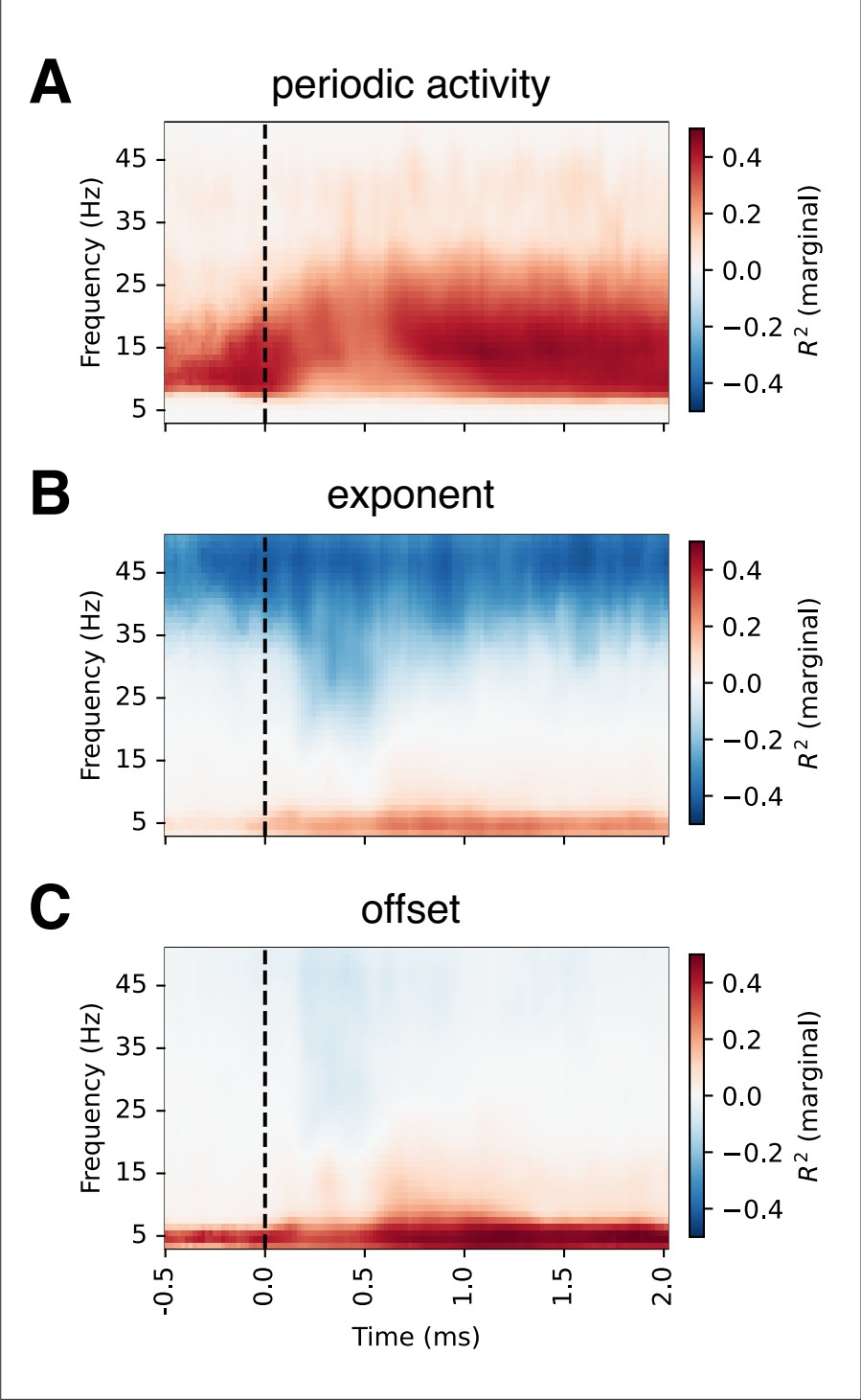

**Figure 4.** Correlations between baseline-corrected time-frequency and FOOOF-decomposed EEG activity. For each FOOOF parameter, we estimated its similarity to baseline-corrected time-frequency EEG activity at each channel-frequency-timepoint using a linear mixed model. We observed strong correlations between periodic activity and baseline-corrected time-frequency activity in the alpha and beta ranges. In contrast, the exponent and offset showed strong correlations in the low-frequency range, with the exponent also showing negative correlations in the gamma range. These results suggest that changes in aperiodic activity appear as low-frequency power in baseline-corrected time-frequency plots. Note that while $R^2$ is strictly non-negative, we assigned its sign based on the beta coefficient from the fixed effect in the model to facilitate interpretation. The values presented

*Figure 4 continued on next page*

*Figure 4 continued*

are averaged across all channels; see *Figure 4—figure supplement 1* for topographical distributions. Condition-specific correlations are shown in *Figure 4—figure supplements 2–4*.

The online version of this article includes the following figure supplement(s) for figure 4:

**Figure supplement 1.** Topographies of correlations between baseline-corrected time-frequency and FOOOF-decomposed EEG activity.

**Figure supplement 2.** Correlations between baseline-corrected time-frequency activity and FOOOF-decomposed activity, separately for each condition, for the exponent.

**Figure supplement 3.** Correlations between baseline-corrected time-frequency activity and FOOOF-decomposed activity, separately for each condition, for the offset.

**Figure supplement 4.** Correlations between baseline-corrected time-frequency activity and FOOOF-decomposed activity, separately for each condition, for the periodic activity.

**Figure supplement 5.** Correlations between baseline-corrected time-frequency and FOOOF-decomposed EEG activity (control dataset).

**Figure supplement 6.** Correlations between baseline-corrected time-frequency and FOOOF-decomposed EEG activity (item-recognition task).

only weak correlations in the low-frequency range. In contrast, both the aperiodic exponent and offset parameters showed strong positive correlations with baseline-corrected power in the theta range. In addition, the exponent showed strong negative correlations with activity above 35 Hz, suggesting that higher frequency power decreases as the aperiodic slope steepens.

## Rhythmicity analysis reveals aperiodic parameters are largely independent of oscillatory activity

A possible concern is that the FOOOF decomposition might not be sufficiently sensitive to peaks at the edges of the frequency spectrum, potentially affecting the interpretation of the results (we thank the reviewers for alerting us to this concern and suggesting an approach to resolve it). To address this issue, we conducted simulations demonstrating that when periodic components include peaks at low frequencies (around 3 Hz), they may go undetected, thereby inflating estimates of aperiodic activity (see the Appendix 1 for details).

To overcome this limitation, we employed an alternative metric to assess oscillatory activity: the recently developed measure of rhythmicity, the phase autocorrelation function (pACF) (*Myrov et al., 2024*). The pACF quantifies the temporal stability of phase dynamics and captures oscillatory activity independent of amplitude fluctuations. Consistent results between FOOOF and pACF would strengthen the conclusion that low-frequency oscillations were truly absent from the recordings, rather than simply undetectable by FOOOF.

As expected, the observed rhythmicity patterns were largely consistent with periodic power, particularly in the alpha frequency range (*Figure 5*, *Figure 5—figure supplement 1*, *Figure 5—figure supplement 2*). In contrast, correlations between aperiodic parameters and pACF were weak in comparison, supporting the notion that the aperiodic parameter estimates remain largely independent of oscillatory processes.

## Periodic activity changes with working memory load, stimulus type, and modality

Next, we examined the differences in periodic activity as a function of task conditions. We observed a decrease in the alpha and beta bands of periodic activity in the 2-back compared to the 0-back condition over the entire trial period and over the entire scalp (*Figures 6 and 7*). Furthermore, we found a greater decrease in alpha and beta power in response to targets compared to non-targets. In addition, we observed a correlation between beta power and reaction times in central channels. This correlation was positive in the early period (up to 0.5 s post-stimulus) and negative in the late period (1–2 s post-stimulus). Finally, we observed an interaction between modality and stimulus type in the time period from 0.5 to 1 s post-stimulus, reflecting a greater difference between target and non-target stimuli in the verbal task in the alpha and beta bands compared to the visuospatial task (*Figure 6—figure supplement 2*).

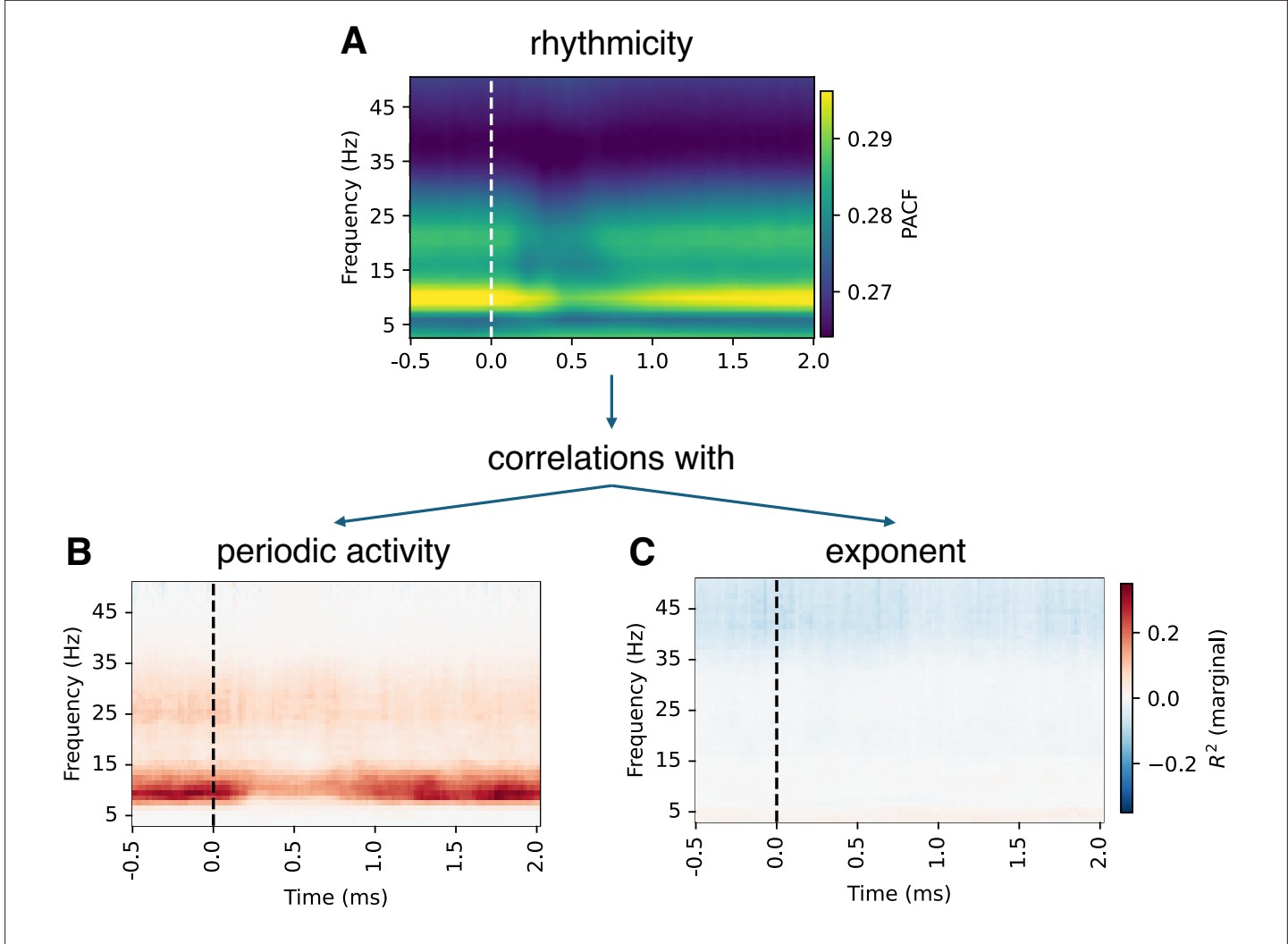

**Figure 5.** Phase-autocorrelation function and its correlations with FOOOF parameters. Phase-autocorrelation function (pACF) (*Myrov et al., 2024*) is an alternative measure of oscillations that is sensitive to rhythmicity rather than the amplitude. Patterns of pACF (**A**) were largely consistent with periodic activity (**B**), particularly in the alpha range, supporting the notion that pACF and periodic activity reflect the same underlying processes. In contrast, the correlations with exponent and offset were low (only results for exponent are shown as results for exponent and offset were essentially the same). See *Figure 5—figure supplement 1* and *Figure 5—figure supplement 2* for detailed figures.

The online version of this article includes the following figure supplement(s) for figure 5:

**Figure supplement 1.** Changes in phase autocorrelation function (index of rhythmicity) as a function of time.

**Figure supplement 2.** Correlations between phase-autocorrelation function and FOOOF parameters.

**Figure supplement 3.** Changes in phase autocorrelation function (index of rhythmicity) as a function of time (control dataset).

**Figure supplement 4.** Correlations between phase-autocorrelation function and FOOOF parameters (control dataset).

**Figure supplement 5.** Changes in phase-autocorrelation function (pACF, index of rhythmicity) as a function of time (item-recognition task).

**Figure supplement 6.** Correlations between phase-autocorrelation function (pACF) and FOOOF parameters (item-recognition task).

## Aperiodic activity has frontal and parietal components

We observed two distinct components (peaks) in the time course of the exponent (*Figure 8*). The first component peaked at around 0.3 s after stimulus onset and was most pronounced in frontal channels, whereas the second component peaked at around 0.7 s post-stimulus and was most pronounced in parietal channels. The exponent exhibited significant differences between stimulus types (*Figure 9*). Specifically, the exponent was larger for targets at approximately 0.3 s and for non-targets at

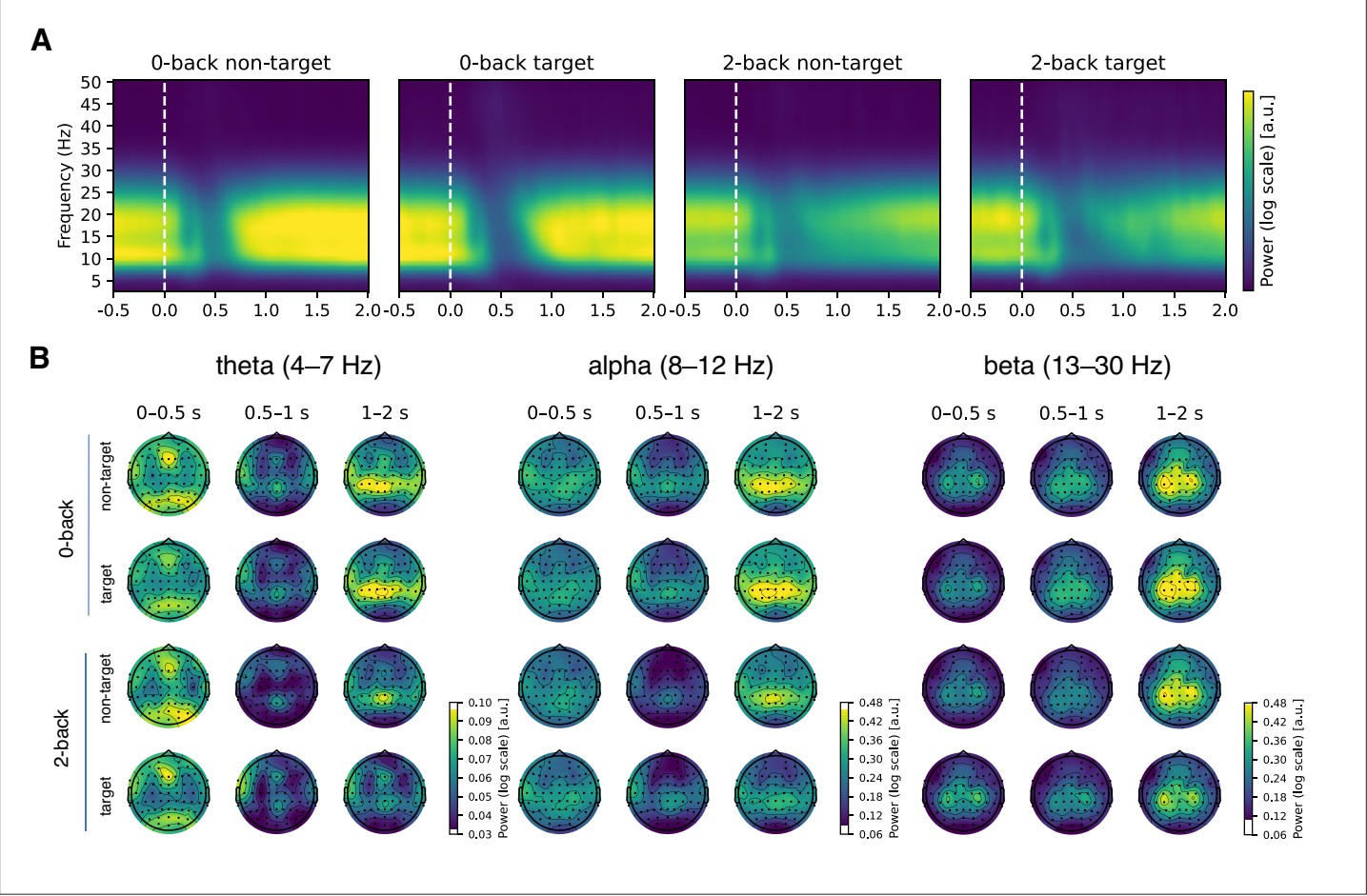

**Figure 6.** Changes in periodic activity as a function of time. (**A**) The inspection of periodic activity revealed strong activity in the alpha and beta frequency bands, with a sharp decrease at around 0.5 s post-stimulus. The power was stronger in the 0-back condition, compared to the 2-back condition (see also *Figure 7*). (**B**) Early beta activity was most prominent at occipital and frontal channels. Note that the colour scale ranges in panel B differ between frequency ranges.

The online version of this article includes the following figure supplement(s) for figure 6:

**Figure supplement 1.** Power spectra of periodic activity on the Fz channel for each participant, averaged over time (between 0.5 and 2 s).

**Figure supplement 2.** Power of periodic activity 0.5 to 1 s post-stimulus, averaged across all channels and participants.

**Figure supplement 3.** Changes in periodic (oscillatory) activity as a function of time.

**Figure supplement 4.** Power spectra of periodic activity on the E15 channel for each participant, averaged over time (control dataset).

**Figure supplement 5.** Changes in periodic (oscillatory) activity as a function of time (control dataset).

**Figure supplement 6.** Periodic activity on the item-recognition task.

**Figure supplement 7.** The power spectra of periodic activity on the Fz channel for each participant in the item-recognition task, averaged over time.

**Figure supplement 8.** Group average power spectra of periodic activity in the item-recognition task, averaged over time.

approximately 0.7 s post-stimulus. We observed no other differences between conditions. We found associations with reaction times at approximately 1.4 s post-stimulus in the parieto-occipital channels.

Similar to the exponent, offset also showed a pattern of activity with two components (see *Figure 8—figure supplement 1*). In addition, offset was sensitive to differences in load (lower offset for 2-back), and there was also an interaction between load × and stimulus type (see *Figure 9—figure supplement 1*).

## Aperiodic activity does not reflect event-related potentials

To ensure that the aperiodic activity identified in our study did not simply reflect event-related potentials (ERPs), we removed evoked activity from the data by subtracting ERPs before performing the

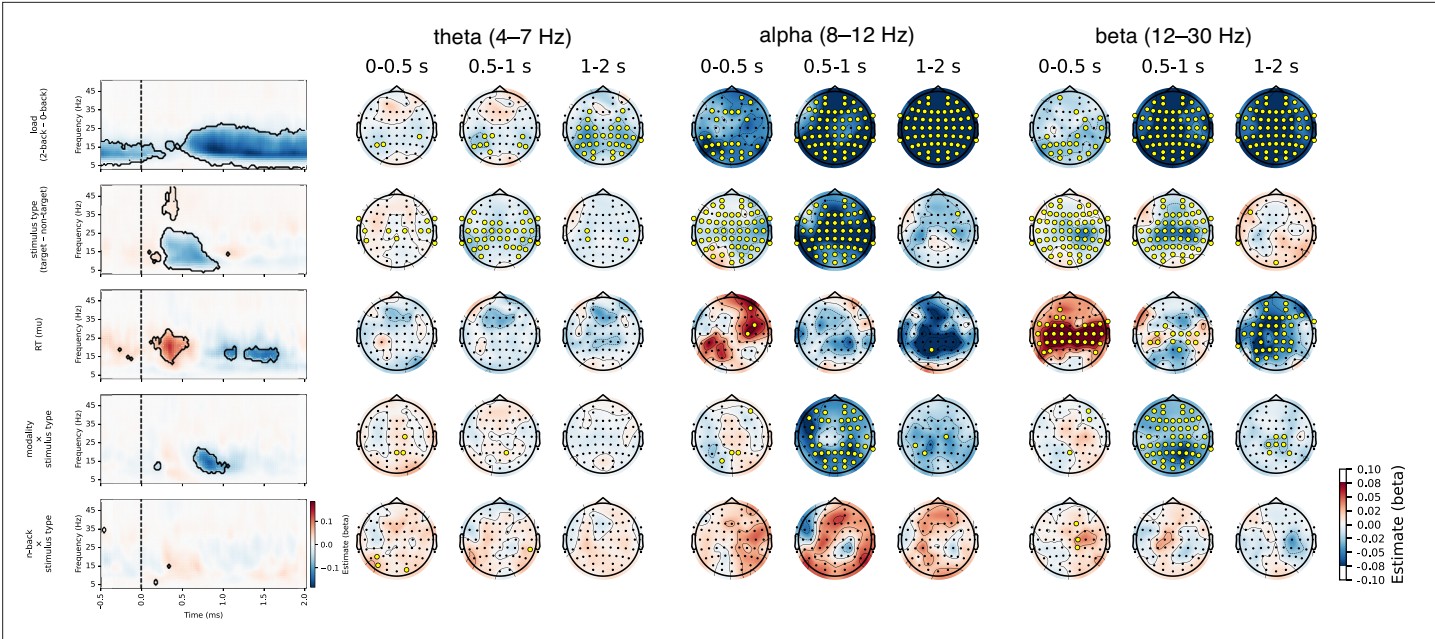

**Figure 7.** Results of the linear mixed model analysis of periodic activity for comparison between conditions. Significant values are highlighted on the heatmap and marked with yellow circles on the topographies. Only factors with significant differences are shown. A p-value was interpreted as significant if it was significant in at least three channels or three time points (see Methods for details). Note that there are small discrepancies between significant values on heatmaps and topographies due to averaging across time or channels. A significant difference was observed between the 2-back and 0-back conditions, with a reduction in activity in the 2-back condition, particularly in the alpha and beta frequency bands, starting at 0.5 s post-stimulus. The differences between stimulus types were most evident from 0.3 to 1 s post-stimulus, with decreased activity for targets compared to non-targets across the whole scalp. Additionally, a smaller effect of the modality × stimulus type interaction was observed from 0.6 to 1 s post-stimulus (see also *Figure 6—figure supplement 2* for detailed visualisation of the interaction). Associations with reaction times were significant in the beta band across central channels. These associations exhibited a positive correlation in the early phase (0–0.5 s) and a negative correlation in the later phase (1–1.5 s).

The online version of this article includes the following figure supplement(s) for figure 7:

**Figure supplement 1.** Results of linear mixed model on periodic activity for comparison between conditions (control dataset).

time-frequency analysis. We then used a linear mixed model to assess the association between the ERPs and aperiodic activity for each channel-time point. We observed $R^2$ values ranging from 0.1 to 0.2 around stimulus onset, but otherwise, correlations were negligible (*Figure 8—figure supplement 4*). This was observed throughout the entire time period examined. In addition, a visual comparison of the topographies of stimulus- and response-locked ERPs with those of aperiodic activity (*Figure 8—figure supplement 3*) revealed no clear similarities, except for a central ERP component observed around 0.25–1 s after the response, which resembled the fronto-central aperiodic component. However, while this component in aperiodic activity showed large differences between targets and non-targets, in ERPs the component differed more between 0- and 2-back conditions.

We also repeated the analysis without subtracting the ERPs (*Figure 6—figure supplement 3*, *Figure 8—figure supplement 5*, *Figure 8—figure supplement 6*), and the results were largely similar. Nevertheless, when ERP was not subtracted, we observed a more pronounced increase in both aperiodic and periodic power between 0 and 0.5 s following stimulus presentation compared to the case with subtracted ERPs.

## Replication confirms main findings

To ensure the generalisability of the findings, we repeated all analyses using two additional datasets. First, we analysed a publicly available dataset from a previously published study by *Nakuci et al., 2023*, in which participants performed a verbal n-back task with three load levels (0-, 1-, 2-back). Next, we analysed the data from an item-recognition task.

In both analyses, we were able to replicate the main findings previously reported, namely the task-related modulation of periodic alpha and beta activity and the task-related changes in aperiodic

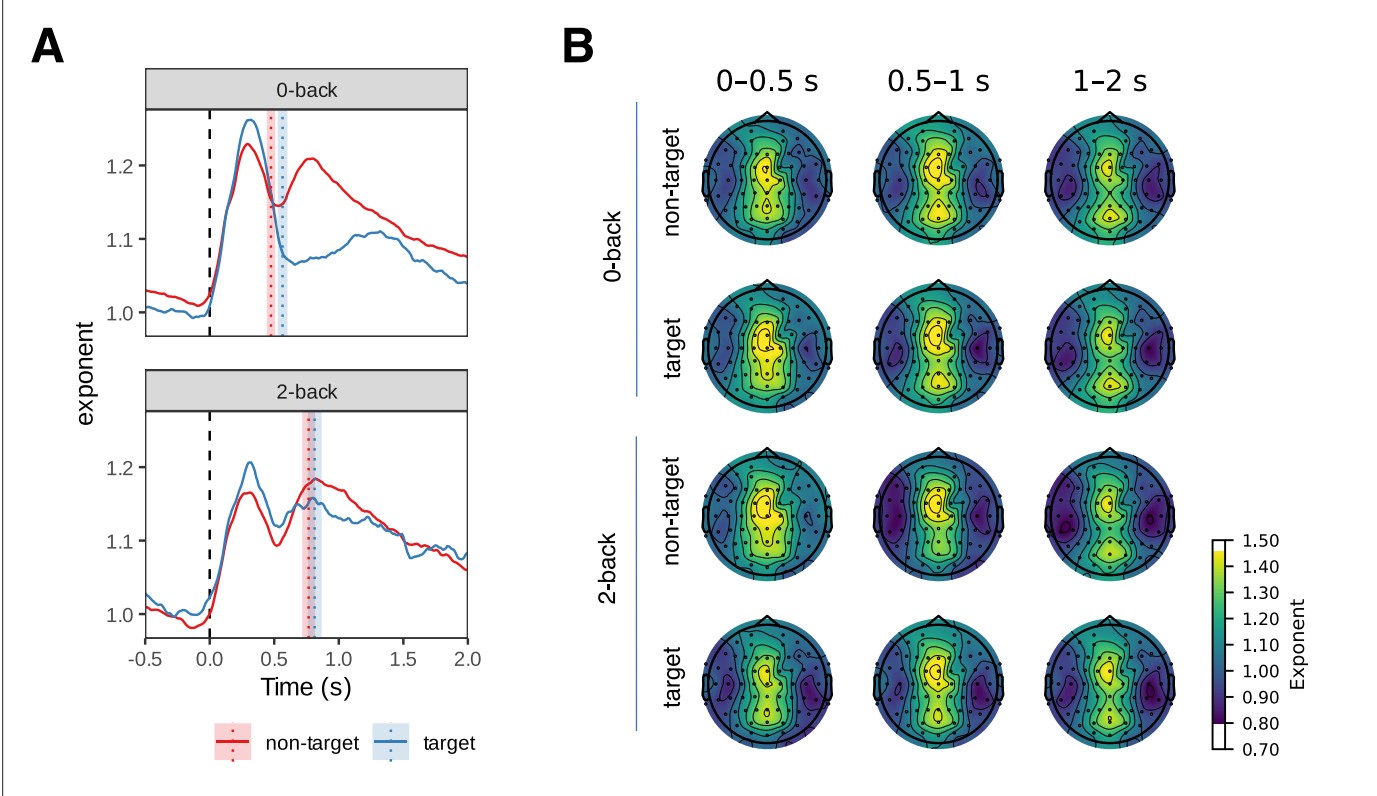

**Figure 8.** Changes in aperiodic activity (exponent, interpreted as aperiodic slope) as a function of time. (**A**) We averaged the time course of the aperiodic activity (exponent) over all channels and observed two components (peaks) of the aperiodic activity. The first component peaked around 0.3 s post-stimulus in the frontal channels. The second component peaked at 0.7 s, was stronger in parietal channels and differed between non-target and target conditions (see *Figure 9*). The time course of the offset parameter was comparable, although the separation between the frontal and parietal components was more pronounced (see *Figure 8—figure supplement 1*). The vertical lines in panel A represent mean reaction time modes. The shaded areas represent 95% Cousineau-Morey within-subjects confidence intervals, adjusted so that non-overlapping intervals correspond to statistically significant differences (*Cousineau, 2017*).

The online version of this article includes the following figure supplement(s) for figure 8:

**Figure supplement 1.** Temporal changes in aperiodic activity (offset).

**Figure supplement 2.** Grand average event-related potentials (ERPs) on midline electrodes.

**Figure supplement 3.** Topographies of stimulus- and response-locked ERPs.

**Figure supplement 4.** Correlations between ERPs and aperiodic activity.

**Figure supplement 5.** Changes in aperiodic activity (slope or exponent) as a function of time.

**Figure supplement 6.** Changes in aperiodic activity (offset) as a function of time.

**Figure supplement 7.** Changes in aperiodic activity (exponent) as a function of time (control dataset).

**Figure supplement 8.** Changes in aperiodic activity (offset) as a function of time (control dataset).

**Figure supplement 9.** Aperiodic activity (exponent) in the item-recognition task.

**Figure supplement 10.** Aperiodic activity (offset) in the item-recognition task.

activity. In the baseline-corrected data, we observed a decrease in alpha and beta power accompanied by an increase in low-frequency power (*Figure 3—figure supplement 3B*, *Figure 10B*). The latter was not evident in the grand average periodic activity (*Figure 3—figure supplement 3C*, *Figure 10C*, *Figure 6—figure supplement 8*) and was only detected in two participants in the n-back task (*Figure 6—figure supplement 4*). In the item-recognition task, a peak in the theta range was present in approximately one-third of the participants (*Figure 6—figure supplement 7*). In half of these cases, the peaks were at 7–8 Hz, which could be considered low alpha peaks rather than theta peaks.

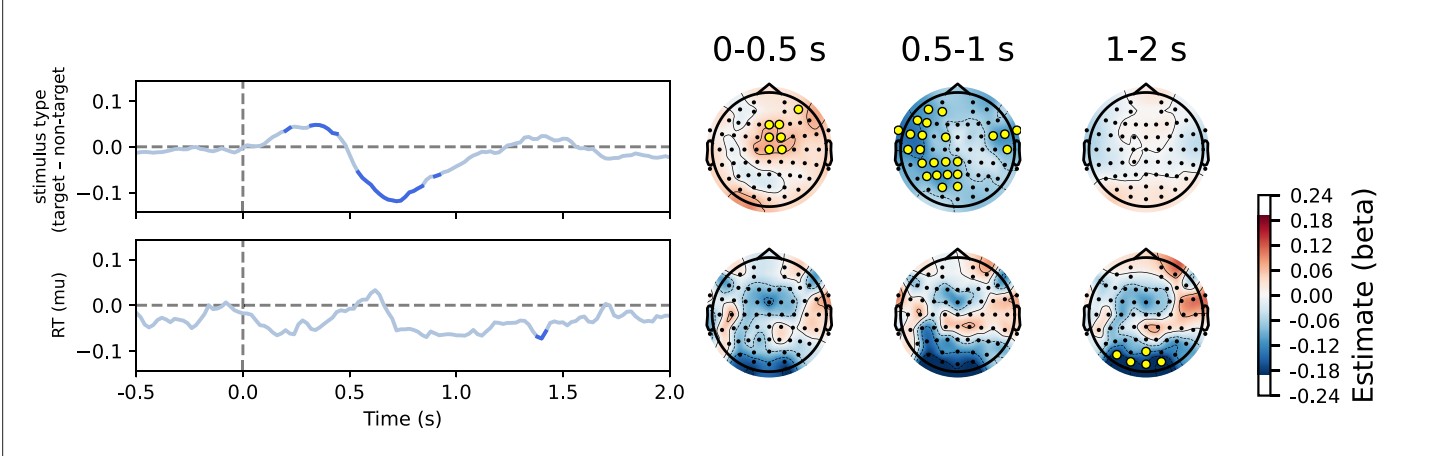

**Figure 9.** Results of the linear mixed model analysis of aperiodic exponent (interpreted as slope) for comparison between conditions. Significant values are shown in blue on line plots and marked with yellow circles on topographies. Only factors with significant differences are shown. The only significant differences between conditions were observed between target and non-target conditions, where the exponent was higher for targets in the early phase (0–0.5 s post-stimulus) and for non-targets in the middle phase (0.5–1 s post-stimulus). There was also a small association with reaction times in occipital channels around 1.5 s post-stimulus. Results were similar for the offset parameter, with an additional effect of load and n-back × stimulus type interaction (*Figure 9—figure supplement 1*).

The online version of this article includes the following figure supplement(s) for figure 9:

**Figure supplement 1.** Results of the linear mixed model on aperiodic activity (offset parameter) of the comparison between conditions.

**Figure supplement 2.** Results of linear mixed model on aperiodic activity (exponent) for comparison between conditions (control dataset).

**Figure supplement 3.** Results of linear mixed model on aperiodic activity (slope) for comparison between conditions (control dataset).

Notably, the correlation patterns between baseline-corrected time-frequency data and FOOOF parameters, as well as between FOOOF parameters and pACF, were consistent across all three datasets. Specifically, aperiodic parameters were associated with low-frequency (theta) baseline-corrected activity, whereas periodic activity correlated with alpha and beta baseline-corrected activity and pACF (*Figure 5—figure supplement 3*, *Figure 4—figure supplement 5*, *Figure 5—figure supplement 4*, *Figure 5—figure supplement 5*, *Figure 4—figure supplement 6*, *Figure 5—figure supplement 6*).

Regarding aperiodic activity, we observed frontal and parietal/occipital components in both tasks. In the item-recognition task, these components were clearly observable only for the offset parameter, whereas for the exponent, the two components were not clearly distinguishable, likely due to lower spatial resolution (32 channels). In both tasks, aperiodic activity was modulated by task demands, with an increase in the exponent following stimuli presentation and response (*Figure 8—figure supplement 7*, *Figure 8—figure supplement 9*).

## Discussion

This study provides a detailed analysis of the relationship between periodic and aperiodic neural activity during the n-back task. EEG data recorded from 57 participants performing both 0-back and 2-back tasks in visuospatial and verbal modalities were analysed to distinguish these two components within the EEG frequency spectrum. The analysis shows the presence of periodic activity in the alpha/beta frequency range prior to stimulus onset, which modulates according to task demands. Furthermore, the results suggest that low-frequency power may largely reflect the effect of baseline correction on aperiodic activity, which also exhibits its own task-related dynamics. This insight is crucial for the interpretation of EEG results, especially for continuous tasks such as the n-back, which require sustained working memory engagement.

In addition to the interplay between the periodic and aperiodic EEG activity, the study expands on the findings of stimulus-induced aperiodic slope shift by identifying two neural components within the aperiodic activity. These aperiodic components are distinguished by their frontal and posterior topographic locations. Additionally, we observed task-dependent differences, including a slightly different slope modulation in non-target compared to target trials. However, we did not find any differences

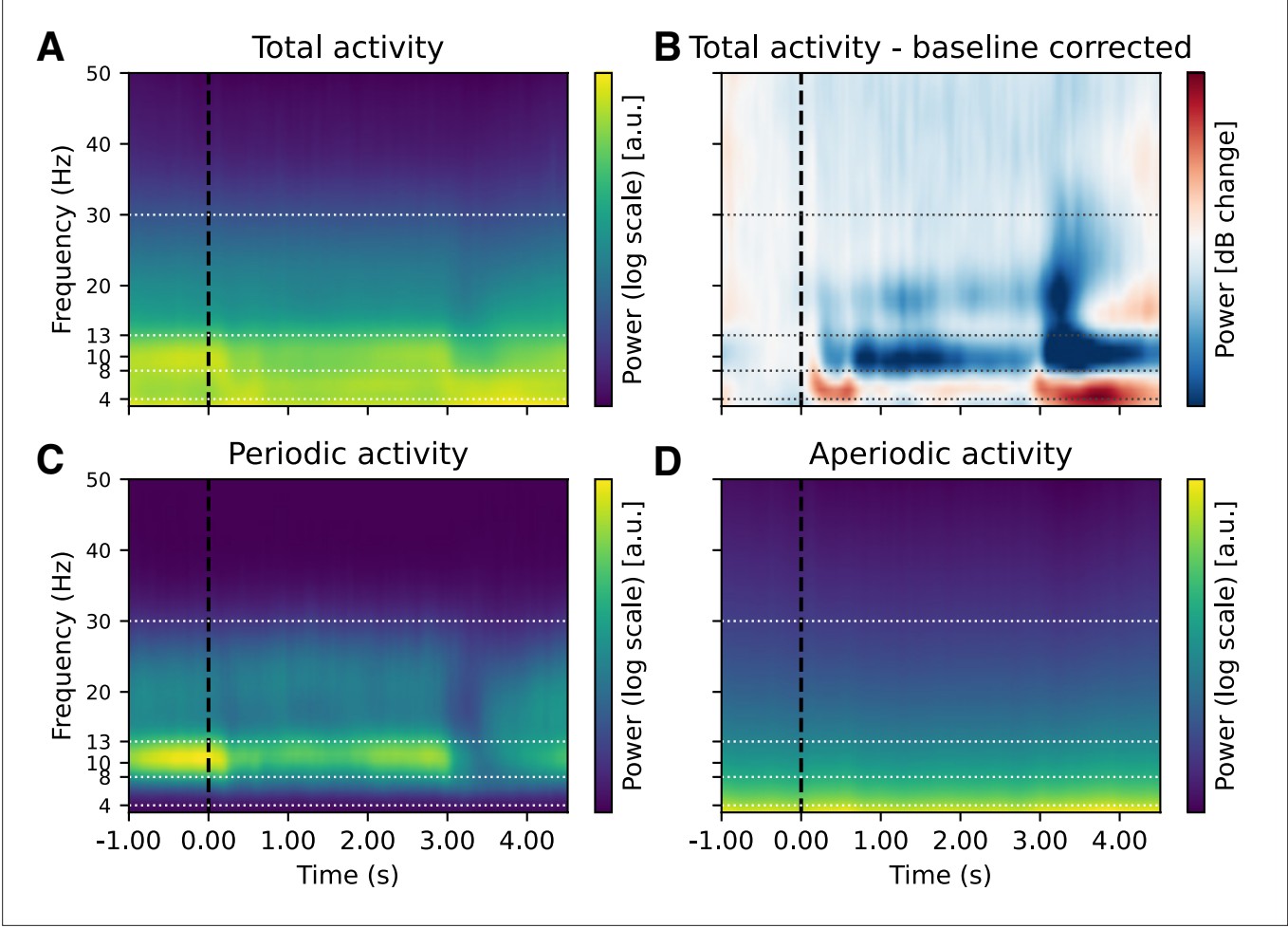

**Figure 10.** The effect of baseline correction and FOOOF decomposition on time-frequency decomposition in the item-recognition task. Similar to the n-back task (***Figure 3***, ***Figure 3—figure supplement 3***), a decrease in alpha power is observed following instruction and stimuli presentation (up to 0.8 s) and continues throughout the retention period, up to 2.8 s post-stimulus (**B**). This is followed by a decrease in alpha and beta power during probe presentation, which is likely indicative of a motor response. Simultaneously, there is an increase in low-frequency power, which is most pronounced during stimulus presentation (up to 0.8 s) and again after probe presentation (after 2.8 ss). The FOOOF decomposition indicates that a substantial portion of low-frequency activity could be attributed to the aperiodic component (**C**, **D**) (see also ***Figure 4—figure supplement 6***). The data shown represent the group average over all conditions at electrode Fz, where low-frequency activity was most pronounced (see also ***Figure 6—figure supplement 8***). Horizontal lines indicate the boundaries of the frequency ranges. See ***Figure 10—figure supplement 2*** for the corresponding figure with a logarithmic y-axis.

The online version of this article includes the following figure supplement(s) for figure 10:

**Figure supplement 1.** Schematic representation of the item-recognition task.

**Figure supplement 2.** The effect of baseline correction and FOOOF decomposition on time-frequency decomposition in the item-recognition task.

between modalities. Collectively, these findings enhance our understanding of the neural dynamics involved in cognitive processes during a continuous working memory task.

## Disentangling periodic and aperiodic components in EEG activity: avoiding baseline correction

The analysis of the EEG data shows that when baseline correction is applied to the total EEG activity, changes in alpha, beta and low-frequency activity are observed following stimulus presentation compared to the pre-stimulus interval (***Figure 3B***). In particular, there is a significant increase in power in the low-frequency range after the stimulus, while a decrease in power in the alpha/beta range is observed for some time after the stimulus presentation. These findings are consistent with previous n-back studies reporting that the presentation of both target and non-target stimuli elicits long-lasting

increases in low-frequency power and decreases in alpha/beta power at stimulus onset (*Palomäki et al., 2012*; *Krause et al., 2010*; *Coleman et al., 2023*).

Omitting baseline correction and decomposing the total EEG activity into periodic and aperiodic components provides a more nuanced insight into the dynamics of the spectra, showing that alpha and beta activities are periodic while low-frequency (delta and theta) activity 'disappears' from the periodic component. These findings suggest that the apparent power in the low-frequency range may, at least in part, reflect changes in the aperiodic component, which could contribute to the post-stimulus increase in low-frequency power observed under baseline correction in time-frequency analysis of the full EEG spectrum. Consistent with this interpretation, we found that aperiodic parameters exhibited the strongest correlations with low-frequency baseline-corrected activity. Given the current understanding of aperiodic activity, specifically based on the findings of post-stimulus slope steepening in the context of increased cognitive demands (*Akbarian et al., 2023a*; *Gyurkovics et al., 2022*; *Virtue-Griffiths et al., 2022*), these results suggest that changes in power within the low-frequency range may be influenced by aperiodic slope modulation rather than a distinct increase in oscillatory low-frequency activity. At the same time, alpha and beta activity remain present as part of the periodic component throughout the task.

In line with this view, the analyses show that low-frequency activities do not necessarily exhibit an oscillatory characteristic during the n-back task (see *Figures 3C, D and 6*). Following a detailed visual inspection at the individual level (*Figure 6—figure supplement 1*), we found that only two participants exhibited evident theta peak, while only one participant displayed pronounced theta and alpha peaks. The majority of participants exhibited alpha peaks, while only two participants displayed a theta peak with no substantial alpha activity. This pattern points to a low-frequency activity that may support the cognitive demands of the n-back task, yet it appears highly variable across individuals, and potentially within individuals, depending on task requirements. A recent study of visual working memory reached a similar conclusion, reporting a robust alpha peak in 90% of participants but inconsistent or absent theta peaks in 60% of participants (*van Engen et al., 2024*).

A potential concern, also raised by reviewers of the original version of this paper, is that FOOOF decomposition may miss low-frequency peaks near the edges of the spectrum. Indeed, simulations we performed showed that in scenarios with low-frequency periodic components, FOOOF may fail to identify these components, leading to inflated estimates of aperiodic parameters (see the Appendix 1 for details). To address this issue, we performed control analyses using the recently introduced measure of rhythmicity, the phase autocorrelation function (pACF) (*Myrov et al., 2024*), which quantifies oscillatory activity by assessing the temporal stability of phase independent of amplitude. This metric showed no evidence of strong low-frequency periodic activity in our dataset, providing further evidence that the observed aperiodic slope reflects genuine non-oscillatory dynamics, rather than an overlooked theta peak. Furthermore, weak correlations between aperiodic parameters and pACF reinforce the independence of these aperiodic features from classical oscillations.

The obtained results show that classical baseline correction can remove continuous oscillatory activity that is present both during baseline and after stimulus onset, because it treats all baseline signals as 'background' to be removed without distinguishing between transient and continuous periodic activity. While this is consistent with the intended purpose of baseline correction–to highlight changes relative to ongoing activity–it may also lead to unintended consequences, such as misinterpreting aperiodic activity as an increase in post-stimulus theta power. Importantly, we confirmed that these observations hold regardless of whether baseline normalisation was performed using a subtractive or divisive approach, suggesting that the conclusions are not driven by any single normalisation strategy.

## The modulation of periodic activity is concentrated in the alpha and beta bands

When decomposing the EEG signal into periodic and aperiodic components, we still observed a significant attenuation in the power of alpha and beta power after stimulus presentation (see *Figures 3B, C , and 6A*). This is in line with existing literature that documents the decrease of post-stimulus alpha and beta (*Krause et al., 2010*; *Coleman et al., 2023*). Moreover, the results support previously reported increased suppressive response that is elicited by target stimuli compared to non-target

stimuli (*Palomäki et al., 2012*; *Peng et al., 2015*), especially in the alpha range (*Figure 6—figure supplement 2*).

Our findings support existing research (*Pesonen et al., 2007*) by demonstrating that reductions in alpha and beta powers are more pronounced and have longer duration under increased task demands, as is typical of higher loads in n-back tasks (see *Figure 7*). Consistent with seminal work (*Gevins et al., 1997*; *Pesonen et al., 2007*; *Krause et al., 2010*), the results of this study show that working memory load mainly affects alpha/beta power in the parietal-occipital electrodes (see *Figure 6B*). In n-back tasks, diminished alpha power is thought to indicate increased engagement in working memory operations (*Haegens et al., 2014*). The functional significance of beta oscillations remains a topic of debate, with studies often linking centrally located beta oscillations to the coordination of motor responses (e.g. *Pavlov and Kotchoubey, 2022*; *Pfurtscheller et al., 1998*; *Kaiser et al., 2001*). However, recent research indicates that they also play a crucial role in working memory, particularly in supporting the maintenance of memorised information and preventing distractions (*Schmidt et al., 2019*; *HajiHosseini et al., 2020*). Additionally, beta oscillations are believed to facilitate the manipulation of stored information, dynamically adjusting to meet task demands and to serve a gating function by inhibiting irrelevant inputs, thus enhancing cognitive efficiency (*ElShafei et al., 2022*; *Schmidt et al., 2019*).

The strong correlation between alpha, lower beta activity and reaction times in central regions, as shown in *Figure 7*, suggests a shared relationship between these measures and shows that beta activity, typically associated with motor function, is most pronounced in central electrodes, although no direct temporal relationship with reaction times was observed. We also observed alpha/beta activities in the parietal electrodes, which support current theories that suggest a collaborative function for alpha and beta oscillations in supporting advanced cognitive functions such as working memory (*Scharinger et al., 2017*; *Coleman et al., 2023*; *Hanslmayr et al., 2016*).

## Aperiodic activity dynamics and their cognitive implications

The analysis of aperiodic activity at each time point within the peristimulus interval builds upon prior research that has documented stimulus-induced changes in the aperiodic component, specifically the post-stimulus steepening of the aperiodic slope (*Gyurkovics et al., 2022*; *Virtue-Griffiths et al., 2022*; *Kałamała et al., 2024*). The results of our study indicate the presence of two components in the dynamics of the aperiodic slope during the n-back task, differentiated both temporally and spatially.

In terms of temporality, the first component appears early in the task cycle, whereas the second component appears at a later stage (see *Figure 8A*). The presence of these components differs between target and non-target stimuli, with a more pronounced difference for the second component (see *Figure 9*). While non-target stimuli consistently exhibit a clear second component, in the case of target stimuli, the slope returns to its baseline steepness more quickly, with a significantly reduced or non-existent second slope steepening. This raises the question of whether the two components represent distinct cognitive operations inherent to the n-back task.

We speculate that the second component, which emerges more prominently following non-target stimuli, may be indicative of an elevated state of vigilance. Since it is rare for one target stimulus to be immediately succeeded by another, participants may become accustomed to a period of 'rest' following a target stimulus, given the low anticipation for another. The increased aperiodic slope following non-target stimuli may reflect the level of alertness for potential targets, whereas after actual target stimuli, vigilance may diminish more rapidly, accounting for the absence of a second slope increase.

The two aperiodic components also exhibit distinct spatial manifestations. The first component is predominantly observed on the frontal electrodes, whereas the second component is also present in the parietal channels. This topographical distinction further supports the hypothesis that these two components reflect different cognitive processes. The activation in the parietal channels may be related to anticipatory mechanisms for the next stimulus, whereas the activation in the frontal channels may be involved in the decision-making process regarding the classification of the previous stimulus as a target.

Overall, the results strongly suggest that solely examining total EEG power or periodic dynamics, without distinctly analysing the aperiodic slope, would overlook a subtle yet potentially significant cognitive processing feature. This feature may be related to distinct working memory processes, which warrants further research.

Importantly, to exclude the possibility that aperiodic activity simply reflects the event-related potentials (ERPs; see *Figure 8—figure supplement 2*), ERPs were subtracted before performing the time-frequency decomposition. We observed low correlations between the ERPs and aperiodic activity (*Figure 8—figure supplement 4*) at the stimulus onset, but no correlations after 0.5 s, suggesting the task demands within the n-back paradigm manifest in aperiodic modulations that are distinct from other stimulus-locked ERPs. While it is possible that response-related ERPs contributed to the second aperiodic component, several observations suggest otherwise: both aperiodic components were present throughout the entire epoch, differences between conditions diverged between ERPs and aperiodic activity (compare *Figure 8* and *Figure 8—figure supplement 3*), and the associations with reaction times were weak. Moreover, the distribution of the response-locked ERP qualitatively resembled the earlier frontocentral aperiodic component more than the later post-response component. Taken together, these findings suggest that ERPs and aperiodic activity capture distinct aspects of neural processing, rather than reflecting the same underlying phenomenon.

While contrasting response-related ERPs with aperiodic components can help address potential confounds, we believe that ERPs are not inherently separate from aperiodic or periodic activity. For example, ERPs may reflect changes in periodic activity, such as phase resets (*Makeig et al., 2002*), or baseline shifts (*Nikulin et al., 2007*). ERPs may also capture aperiodic activity, either in the form of evoked transients triggered by an event (*Shah et al., 2004*) or induced changes in the ongoing background signal. This distinction has important implications: evoked transients can alter the broadband spectrum without implying shifts in ongoing background activity, whereas induced aperiodic changes may signal different neural mechanisms, such as shifts in the excitation–inhibition balance (*Gao et al., 2017*). Therefore, different approaches to studying EEG activity should be seen as providing complementary rather than competing perspectives.

These findings are consistent with previous research that has consistently observed systematic, stimulus-induced changes in the aperiodic component that are distinct from concurrent stimulus-evoked EEG activity (*Gyurkovics et al., 2022*; *Kałamała et al., 2024*; *Virtue-Griffiths et al., 2022*). We thus underscore the importance of acknowledging the unique characteristics of aperiodic activity, particularly in the context of the n-back task.

## Distinct functions of periodic and aperiodic EEG components during cognitive engagement

The findings on periodic and aperiodic EEG activities during the n-back task suggest a complex pattern of cognitive engagement. Following stimulus presentation, the analysis shows an almost immediate aperiodic slope increase in the fronto-central channels (see *Figure 8*), followed by a brief alpha/beta attenuation in the posterior channels (see *Figure 6*). Alpha activity has been linked to inhibitory processes; accordingly, posterior alpha (with concomitant beta) attenuation is commonly interpreted as a transient release of inhibition that facilitates sensory processing (*Bonnefond and Jensen, 2012*; *Waldhauser et al., 2012*; *Wianda and Ross, 2019*; *Tuladhar et al., 2007*). In contrast, the fronto-central increase in the aperiodic slope is compatible with a brief shift towards stronger inhibitory tone (reduced E:I) following stimulus onset (*Gyurkovics et al., 2022*; *Kałamała et al., 2024*; *Gevins et al., 1997*; *Pesonen et al., 2007*). The results show that both the alpha/beta suppression and the aperiodic change in slope are more pronounced after target stimuli compared to non-target stimuli (see *Figures 7 and 9*). These observations may indicate greater cognitive engagement associated with decision-making processes following the onset of the target stimulus.

Further analysis of the data shows a significant difference between the 0-back and 2-back conditions in the periodic component (*Figure 7*), indicating more sustained alpha/beta suppression in the 2-back condition, whereas the aperiodic component did not show such spectral slope differences. Sustained suppression of periodic alpha and beta activity during the 2-back condition may reflect ongoing active engagement in the form of information encoding, retention, and updating. This finding suggests that the periodic and aperiodic components may reflect distinct cognitive processes. The characteristics of the second component in the aperiodic slope after non-target stimuli (see *Figure 8*), with no obvious co-occurring processes in the periodic component, further support this idea. It is important to analyse both components to ensure that the processes they reflect are not overlooked.

The results of the study show a clear distinction between the periodic and aperiodic components in relation to reaction times. These results support the association between central beta activity and

motor planning (e.g. *Eggermont, 2021*; *Pfurtscheller et al., 1998*), as evidenced by its correlation with reaction times in the analysis of the periodic component (*Figure 7*). In contrast, the analysis indicates a relatively weak correlation between the aperiodic slope and reaction times (*Figure 9*). This observation differs from the most commonly reported patterns (e.g. *Voytek et al., 2015*; *Thuwal et al., 2021*; *Akbarian et al., 2023a*), but the evidence remains inconclusive, as some studies have reported contrasting results (e.g. *Monchy et al., 2023*). The discrepancy highlights the need for further research to understand how different factors influence the manifestation of the aperiodic component as a reflection of cognitive processes.

Prior studies have consistently reported sustained elevations in theta power during the n-back task (e.g. *Palomäki et al., 2012*; *Krause et al., 2010*; *Pesonen et al., 2007*). These increases, particularly in the frontal cortex, have been linked to various cognitive functions such as attention, information encoding, and cognitive control (*Roux and Uhlhaas, 2014*; *Rossi et al., 2023*; *Cavanagh and Frank, 2014*). While power in the theta range can indeed be oscillatory, as demonstrated in some studies, the sustained low-frequency activity observed in the analysis of scalp EEG data in this study appears to be largely influenced by aperiodic components, with post-stimulus increases potentially attributable to shifts in spectral slope rather than exclusively changes in periodic activity. Because low-frequency activity can reflect either periodic or aperiodic processes, distinguishing between them is key to avoiding misinterpretation. Nevertheless, previously reported patterns of theta modulation are consistent with the notion that post-stimulus changes in aperiodic activity may reflect enhanced inhibitory control commensurate with task demands (*Voytek et al., 2015*; *Gao et al., 2017*).

## Main findings replicated in additional datasets

Analyses of control data from a previously published study employing n-back task (*Nakuci et al., 2023*) confirmed our initial results, indicating that similar task-related modulations in alpha and beta periodic activities, as well as aperiodic activity dynamics, can be divided into frontal and parietal components. However, although both aperiodic components were observed in the control dataset, their temporal differentiation was less pronounced (*Figure 8—figure supplement 7*, *Figure 8—figure supplement 8*). Notably, in this comparative dataset, the aperiodic slope was only related to task load, with a steeper slope observed for the 2-back level, and no significant differences based on stimulus type (*Figure 9—figure supplement 2*). Furthermore, in the comparative dataset, there was an association with reaction times in the parietal channels, more so than in our original dataset.

To test whether these findings generalise across different task paradigms, we applied the same methods to an item-recognition task, which showed similar low-frequency dynamics despite having distinct mechanisms of memory maintenance and a non-continuous structure (*Figure 10*). Analyses of the original n-back dataset, the control n-back dataset, and the item-recognition dataset all point to the same core pattern: low-frequency power in baseline-corrected data correlates with aperiodic activity, while periodic activity is associated with alpha/beta power in both baseline-corrected data and pACF. This cross-task consistency underscores the robustness of the interpretation that periodic and aperiodic components should be carefully distinguished when studying working memory.

## Theoretical and methodological implications

This study highlights the importance of refined EEG analysis methods that take into account the dynamic nature of both periodic and aperiodic components during cognitive tasks. The findings suggest a need for reconsidering baseline correction techniques, particularly in continuous paradigms where traditional baseline subtraction can obscure meaningful neural dynamics by potentially misrepresenting aperiodic activity as low-frequency (theta) power (*Gyurkovics et al., 2021*). Such misrepresentation is particularly relevant under sustained cognitive demand, as corrections intended to normalise data may inadvertently lead to misleading or artefactual interpretations of neural activity. Further investigation is warranted to explore how changes in the aperiodic slope might underlie the commonly observed increases in frontal low-frequency power, which have traditionally been referred to as theta oscillations. In order to further validate this interpretation, future research should extend to a wider range of working memory and cognitive control tasks.

The aperiodic slope dynamics observed during the n-back task in this study align with the E:I balance framework (*Gao et al., 2017*; *Vogels and Abbott, 2009*; *Lim and Goldman, 2013*). Specifically, stimulus-induced shifts in the slope likely reflect a proportional modulation of ongoing excitatory

activities to meet processing demands (*Gyurkovics et al., 2022*; *Voytek et al., 2015*; *Virtue-Griffiths et al., 2022*), consistent with a transient increase in inhibitory control that relates to frontal theta indices of cognitive control (e.g. *Scharinger et al., 2017*; *Cavanagh and Frank, 2014*; *Ratcliffe et al., 2022*). Building on this account, lifespan studies show that the aperiodic exponent at rest becomes flatter with advancing age. This is often described as greater neural noise, and this shift relates to lower cognitive performance and prospective decline (*Voytek et al., 2015*; *Cesnaite et al., 2023*; *Finley et al., 2024*). In this context, age is a plausible moderator of task-evoked slope dynamics in the n-back task, with older adults expected to show reduced or delayed steepening. This is consistent with event-related evidence regarding aperiodic modulation and resting-state links to cognition (*Kałamała et al., 2024*; *Finley et al., 2024*; *Cesnaite et al., 2023*; *Dave et al., 2018*). As our main sample did not consist of young adults, it is possible that age moderates these aperiodic slope effects. However, as the present study was not designed or powered to test for interactions between age and condition, we treat this as a theoretical implication to be evaluated in future work involving a broader age range. Taken together, these observations suggest that aperiodic slope modulations should be considered a potential biomarker of cognitive control, and that their relationship with established measures should be investigated to predict performance across tasks.

Furthermore, investigating the potential relationship between aperiodic slope dynamics and theta oscillations in tasks that require enhanced cognitive control could provide a deeper understanding of the neural basis of cognitive flexibility and decision-making. Considering individual variations in theta dynamics and their relationship with aperiodic slope modulation is also important, as this could reveal potential biomarkers for cognitive efficiency or susceptibility to cognitive fatigue.

To better understand the functional role of the second aperiodic component observed after non-target trials (second slope steepening), future work should use a design with a larger dataset and a greater number of sequences in which a target is followed by another target. This would enable a detailed comparison of aperiodic components in scenarios where a target is succeeded by another target versus when it is followed by a non-target, potentially revealing how attentional mechanisms are reset or sustained under different conditions.

Finally, the findings highlight a potential issue for EEG-based cross-frequency coupling analyses, particularly phase-amplitude coupling (*Tort et al., 2010*; *van der Meij et al., 2012*; *Tseng et al., 2019*). Numerous studies have demonstrated gamma amplitude modulation by the phase of theta (e.g. *Park et al., 2013*; *Köster et al., 2014*; *Goodman et al., 2018*), posited to organise information in working memory (e.g. *Rajji et al., 2016*; *Brooks et al., 2020*; *McGill and Kieffaber, 2024*; *Goodman et al., 2018*). However, the results of this study suggest that scalp-recorded low-frequency fluctuations may be predominantly influenced by aperiodic shifts rather than true oscillations. Consequently, it may be unwarranted to assume an inherently oscillatory theta band when investigating theta-gamma coupling. In future cross-frequency coupling studies, it is, therefore, crucial to confirm that theta power reflects genuine oscillations, rather than aperiodic activity, before drawing conclusions about cross-frequency interactions.

## Strengths and limitations

The main strength of the study is the replication of key findings by performing the same analyses on two independent samples. Replication of the results in an independent sample of younger individuals performing a standard verbal n-back task extends the generalisability of the findings beyond the original cohort of older adults. Furthermore, replication of these results in the item-recognition task demonstrates the robustness of conclusions across different task paradigms.

Furthermore, we used pACF, an alternative measure of oscillations, to verify the robustness of the approach, thereby ensuring that low-frequency oscillatory activity was not simply missed by FOOOF near the edges of the spectrum.

There are a number of limitations that need to be considered. First, the task difficulty in the 2-back condition may have been too high. Some participants had to be excluded from the analysis due to poor performance. This exclusion may have reduced the range of cognitive abilities in the sample of older adults, potentially inflating them.

In addition, the higher level of difficulty in the tasks may have influenced the observed differences between the 0-back and 2-back conditions, potentially affecting the magnitude and direction of the reported neural modulations. For the sample of older adults, a simplified 2-back task using a single

modality may have been more appropriate. The single modality (verbal, with letters displayed in the centre of the screen) may help to reduce task complexity and align better with the cognitive abilities of older participants. In addition, the dual-modality task imposed time constraints that limited the number of trials for each condition, potentially leading to participant fatigue and reduced data quality.

## Conclusions

The study highlights the critical importance of distinguishing between the periodic and aperiodic components of the EEG spectrum in order to ensure accurate interpretation of changes traditionally attributed to oscillatory activity. Using the n-back task, we have demonstrated that the aperiodic component, which is often dismissed as mere background noise, reflects important features of cognitive processing. The findings also challenge the routine practice of baseline correction, which can obscure crucial modulations of aperiodic activity, particularly in continuous cognitive tasks. It is important to note that a substantial proportion of the low-frequency activity commonly observed in scalp EEG may in fact be attributable to shifts in the aperiodic slope. It is, therefore, crucial to independently verify whether the observed low-frequency signals are genuinely periodic (oscillatory) or primarily aperiodic. This is of particular importance for accurately deciphering the neural underpinnings of working memory and cognitive control and highlights the need for a refined approach in the analysis and interpretation of EEG data.

## Methods

### Participants

We analysed EEG recordings from fifty-seven volunteers (nine males) aged 55–78 years (M=67.4, SD = 6.4), who initially participated in a study on subjective cognitive complaints. All participants had normal or corrected-to-normal vision and were free from neurological or psychiatric illness or cognitive decline, as assessed by neuropsychological testing prior to study entry. Of these participants, twenty-seven reported subjective cognitive complaints, while thirty did not. The current study focused exclusively on within-subject comparisons and did not investigate subjective cognitive complaints or between-group differences. We excluded data from 10 participants from all analyses due to poor quality.

The study protocol was approved by the Medical Ethics Committee of the Republic of Slovenia (protocol number 0120-128/2019/9; 21.08.2019). All study participants gave written informed consent prior to the initiation of any study-related procedures.

### Experimental task and procedure

The participants were instructed to press the left button when the letter or position matched the letter/position n-back trials before (i.e. on target trials), and the right button otherwise (on non-target trials). Some of the stimuli were lures, i.e., stimuli that would have been considered targets in the 1-back condition but appeared during the 2-back condition.

The participants completed the n-back task in two modalities (visuospatial and verbal) with two load conditions (0- and 2-back) (*Figure 1*). During the task, participants were presented with white capital letters on a grey background one at a time in one of nine possible positions, in a continuous sequence. Each letter was presented on the screen for 250 ms, followed by a 3000 ms interval during which participants were required to respond. In the verbal task conditions, participants were instructed to memorise the presented letter, whereas in the visuospatial task conditions, participants were asked to memorise the position of the presented letter. The order of modality was counterbalanced across participants.

Each modality always began with the 0-back load condition in which the target stimulus was presented for 10 s at the beginning of the block and stayed the same throughout the whole block. The 0-back load condition was followed by a 2-back load condition of the same modality, where participants had to remember each stimulus as it appeared and discern whether it matched the one displayed two trials prior. The 0-back and 2-back load conditions each comprised three 5 min blocks of continuously presented stimuli. For each modality, the 0-back load condition comprised a total of 201 stimuli, 39 of which were targets, and the 2-back load condition comprised a total of 240 stimuli, 39 of which were targets, and 30 were lures. All stimuli were presented in a pseudorandomised order.

The task began only after participants indicated their readiness by pressing a designated key, after reviewing the instructions and clarifying any doubts with the experimenter. Before each condition, participants completed a practice session consisting of a short block to familiarise themselves with the task demands. The session included a variation of the main task in which participants practised identifying both the target letter and its location, as well as responding to the 2-back challenge for both letters and positions. During the main task, response accuracy and reaction times were recorded by keystroke on the Cedrus response pad (Cedrus Corporation, San Pedro, CA, USA, model RB-540). Participants were instructed to favour accurate responses over fast responses. The participants' responses were monitored, and only trials with correct responses were included in the EEG analysis.

The stimuli were presented in Helvetica font on a 24-inch LCD screen with a refresh rate of 120 Hz. The stimuli, arranged in a 3×3 matrix, spanned approximately 7.22 degrees of both vertical and horizontal visual angles. The task was programmed and executed using PsychoPy 3 (*Peirce et al., 2022*).

## EEG data acquisition

We recorded EEG activity using either a BrainAmp MRplus or ActiChamp amplifier (Brain Products GmbH, Germany). We acquired the recordings with a 64-channel ActiCap plus a reference channel, configured according to the extended 10–10 international system layout. To accommodate all participants, we used two slightly different electrode layouts, differing only in the position of two electrodes. Due to inconsistencies in electrode placement between subjects, we excluded channels AFz, Iz, PO9, and PO10 from the analysis, resulting in a dataset of 63 electrodes (including the reference channel). We grounded the electrode setup at AFz, with FCz serving as the reference. During the EEG recording, we band-pass filtered the signal between 0.1 and 250 (or 1000) Hz with a slope of 12 dB/octave and digitised it at a sampling rate of 500, 1000, or 2500 Hz. After time-frequency decomposition, we downsampled the data to 100 Hz.

## EEG data preprocessing

We preprocessed the EEG data using custom MATLAB scripts (MathWorks, USA) that integrated functions from EEGLAB (*Delorme and Makeig, 2004*) and ERPLAB (*Lopez-Calderon and Luck, 2014*). High-pass filtering at 0.1 Hz was performed using a finite impulse response (FIR) filter (slope: 12 dB/octave). A CleanLine filter was then used to reduce power line noise at 50 Hz (*Mullen, 2012*). Next, we segmented the data into epochs ranging from –1.5–2.8 s, time-locked to stimulus presentations, followed by baseline correction to remove residual signal drift. Epochs and channels were manually inspected to identify and exclude segments with excessive artefacts. Independent component analysis decomposition was performed using the AMICA algorithm (*Palmer et al., 2011*), applied to the data bandpass filtered between 1 and 60 Hz; the solution was later applied to the non-bandpass filtered data. We manually inspected the independent components to identify and remove those representing blinks, horizontal eye movements, muscle noise, line noise, and other artefacts. After the ICA, channel time series reconstruction and channel interpolation were performed, and the data were referenced to the average reference.

Subsequent EEG analyses were conducted only on subjects and conditions that met the required standards, defined as having a minimum of 30 good epochs after cleaning and an accuracy rate of at least 0.6 within a given condition. Exclusion criteria were applied at the level of individual conditions within subjects, rather than excluding entire participants from the analyses. Consequently, only 21% and 10% of participants remained for the visuospatial and verbal lure conditions, respectively. As a result, the lure condition was excluded from all subsequent analyses.

## Estimation of periodic and aperiodic component

First, we excluded trials with incorrect or missing responses from all subsequent analyses. ERPs related to early sensory potentials can affect estimates of aperiodic activity (*Gyurkovics et al., 2022*); therefore, average ERPs were removed prior to time-frequency decomposition, as has been previously done in similar studies (*Gyurkovics et al., 2022*; *Kałamała et al., 2024*). This involved computing ERPs for each subject/condition and then subtracting these ERPs from single-trial data for each subject/condition. For completeness, we also analysed the data without subtracted ERPs.

To estimate both periodic and aperiodic components of EEG activity, we performed time-frequency decomposition using superlets, with a range of frequencies from 3 to 50 Hz, in steps of 1 Hz (width

of the base wavelet: three cycles, order: linearly spaced from 1 to 20, multiplicative combination of wavelets) as implemented in Fieldtrip (*Oostenveld et al., 2011*). Superlets, a variant of the Morlet wavelet transformation, involve multiple decompositions with varying cycles, the results of which are combined using a geometric mean to achieve enhanced temporal and spectral resolution, known as 'super-resolution' (*Moca et al., 2021*). The results of the time-frequency decomposition were averaged across trials for each condition and participant and then decimated to 100 Hz in order to save space and facilitate further analysis.

We then estimated the periodic and aperiodic components from the decomposed data for each time point, condition, and participant using the FOOOF algorithm from the Python specparam package (version 2.0.0rc2) (*Donoghue et al., 2020b*). To avoid edge artefacts, FOOOF parameters were estimated for a period between –0.5 and 2 s around the stimulus. The FOOOF algorithm parameterises the frequency spectrum in a semi-logarithmic space, where only power values are log-transformed, using a set of parameters separately for periodic and aperiodic activity. Specifically, neural power spectra (NPS) over a set of frequencies (F) are described as the sum of the aperiodic and periodic components:

$$NPS(F) = L(F) + G(F)_n \tag{1}$$

The aperiodic component is modelled as

$$L(F) = b - \log\left(k + F^{\chi}\right) \tag{2}$$

where $b$ is the offset, $k$ is the knee (bend), and $\chi$ is the aperiodic exponent (sometimes referred to as the spectral slope).

The periodic component is modelled as a sum of Gaussians:

$$G(F)_n = a * \exp\left(\frac{-(F - c)^2}{2 * w^2}\right) \tag{3}$$

Each Gaussian is described by three parameters: $c$, $a$, and $w$, representing center frequency, power, and bandwidth, respectively.

Fitting was performed using default specparam parameters: peak width limits = 0.5–12.0 Hz, maximum number of peaks to fit = Inf, minimum peak height = 0, peak threshold = 2.0, aperiodic mode = 'fixed' (i.e. the knee parameter was set to 0). Note that the FOOOF fitting was performed on trial-averaged data to increase the stability of the parameter estimates. In contrast, *Gyurkovics et al., 2022*, who also investigated task-related changes in aperiodic activity, stabilised their fits by estimating power spectra over 1 s time windows at the single-trial level. Although effective, this strategy involved a trade-off in temporal resolution that our trial-averaging approach seeks to avoid.

To assess the fit of the FOOOF models, we examined $R^2$ and model error metrics (Appendix 2). Model error was defined as the absolute difference between predicted and actual values. However, as we have shown in the simulation (see Appendix 1), model fit metrics can be misleading. By simulating data and estimating parameters, we observed cases where the model fit was high, but the estimated parameters deviated substantially from the ground truth values. This highlights the limitations of relying solely on fit metrics to assess parameter accuracy.

In addition, we examined the number of peaks identified per model. On average, 1.9–2 peaks were identified, showing high consistency and indicating that the models did not overfit by detecting an unrealistic number of peaks (*Appendix 2—figure 2*). We also repeated the analysis with the maximum number of peaks set to three, which gave virtually identical results (results not shown).

The outcomes of the analyses conducted with the offset and exponent parameters were found to be highly comparable. Consequently, the main text only includes analyses using the exponent parameter, whereas the Supplement contains analyses using the offset parameter.

To allow comparisons with full-time frequency power and to facilitate comparisons across participants and conditions, we reconstructed the periodic component in a semi-log space based on the estimated parameters for each participant and condition using the formula in *Equation 3*. This reconstruction includes only the parameterised spectral peaks and is referred to as 'periodic power' or 'periodic activity,' a measure we used in all subsequent analyses. Importantly, the reconstructed periodic components were strictly non-negative: although Gaussians asymptotically approach zero, they never reach it, resulting in smooth transitions without explicit zeros between peaks.

We visually assessed the time-frequency plots, both in their original form and after baseline correction, before performing formal statistical analyses. To assess the effect of baseline correction on the results, we compared several baseline correction methods, including: (a) decibel conversion ($10 * \log_{10}(data/baseline)$), (b) relative change (($data - baseline)/baseline$), (c) normalised change (($data - baseline)/(data + baseline)$), (d) absolute change ($data - baseline$). In addition, we considered three baseline periods (relative to stimulus onset): (a) from –500 to –200 ms, (b) from –300–0 ms, (b) from –500–0 ms.

## Comparison of baseline-corrected data and FOOOF parameters

We compared baseline-corrected time-frequency activity with both periodic and aperiodic components. To better understand their relationship, we first visually inspected grand-average plots and then performed mass univariate comparisons. Specifically, we examined the relationship between baseline-corrected power and FOOOF parameters for each channel, time, and frequency point.

A simple approach to assessing this relationship would be to compute correlations across subjects at each point. However, this approach would violate the independence assumption of simple linear models, as each subject contributed multiple observations across different experimental conditions. A more appropriate approach was to use a linear mixed model that included both subject and condition as random effects.

Although the condition is typically treated as a fixed effect, in this case, we were not interested in the specific conditions themselves. Instead, we treated them as a random sample from a larger population of possible conditions. Baseline-corrected activity was treated as the dependent variable, while periodic or aperiodic activity served as a fixed effect predictor variable.

For each test, we estimated the marginal $R^2$ (**Nakagawa and Schielzeth, 2013**), a measure of variance explained for linear mixed models, analogous to $R^2$ in general linear models. The marginal $R^2$ is defined as variance explained by the fixed factors using the equation:

$$R^2_{\text{LMM}(m)} = \frac{\sigma_f{}^2}{\sigma_f{}^2 + \sigma_\gamma^2 + \sigma_\alpha^2 + \sigma_\varepsilon^2} \tag{4}$$

where $\sigma_f^2$ represents the variance explained by the fixed effects, $\sigma_\gamma^2$ and $\sigma_\alpha^2$ correspond to variance of random effects (subject- and condition-related variance, respectively), and $\sigma_\epsilon^2$ represents the residual variance. The fixed-effects variance $\sigma_f^2$ can be computed as the variance of the predicted values.

Although $R^2$ is strictly non-negative, we assigned the sign based on the beta coefficient from the fixed effect in the mixed model.

Additionally, to assess variability across participants, we visualised time-averaged periodic activity for each participant (**Figure 6—figure supplement 1**).

## Validation of FOOOF results

As reviewers of the original manuscript raised concerns that FOOOF might miss peaks near the edges of the spectrum (**Gerster et al., 2022**), we performed two control analyses to test the validity of the FOOOF results.

First, we assessed whether the FOOOF decomposition can accurately resolve low-frequency periodic activity by simulating ground truth data with systematically varied parameter values. Details of this simulation are provided in the Appendix 1. Briefly, we found that low-frequency periodic components were often conflated with aperiodic components, resulting in undetected periodic activity at very low frequencies. This issue was more pronounced when fitting was restricted to frequencies $\geq 3$ Hz (as in our analyses of experimental data). In addition, although model fit metrics were generally good, they did not always indicate accurate parameter estimation. These results suggest that even when true low-frequency periodic activity is present, FOOOF may fail to detect it. Conversely, changes in the aperiodic parameters estimates may actually be a consequence of low-frequency periodic activity. Importantly, as noted by the reviewer, this behaviour reflects an explicit design choice of the algorithm: to avoid overfitting ambiguous peaks at the edges of the spectrum, FOOOF excludes peaks that are too close to the boundaries. This exclusion is controlled by the _bw_std_edge parameter, which defines the distance that a peak must be from the edge in order to be retained (in units of standard deviation; set to 1.0 by default). Therefore, although the algorithm is functioning as

intended, users should exercise caution when interpreting aperiodic parameters in datasets where low-frequency periodic activity might be expected.

Second, we used an alternative method to assess whether signals at different frequencies were oscillatory. Specifically, we used the recently developed phase-autocorrelation function (pACF) (*Myrov et al., 2024*). Unlike amplitude-based methods such as FOOOF, pACF is sensitive to the 'rhythmicity' or temporal stability of the phase of a signal. pACF achieves this by calculating the cross-spectrum between a complex signal and its time-delayed copy, then normalising to unit magnitude to remove amplitude information. Because pACF isolates phase relationships while being inherently independent of amplitude, it serves as a complementary measure to FOOOF. Consistent results between FOOOF and pACF would strengthen our conclusion that low-frequency (theta) oscillations were truly absent from our recordings, rather than simply undetectable by FOOOF.

Sample pACF is estimated using the formula:

$$\overline{\mathrm{pACF}}(l) = \frac{1}{N} \sum_{i=1}^{N} \mathrm{CS}_{x,x}(l) \tag{5}$$

where $CS_{x,x}(l)$ represents a normalised cross-spectrum between a complex signal $X$ and its delayed copy at lag $l$, and $N$ is the total number of samples.

To allow comparison with the time-resolved FOOOF decomposition, the time-resolved pACF was estimated as follows:

$$x_t = \sum_{l=0}^{L_{\max}} \frac{\sum_{i=t-W/2}^{t+W/2} z_i^l}{W} \Big/ N \tag{6}$$

where $t$ is a timestep, $l$ is a lag, $W$ is a window size (in samples), $N$ is the number of lags, and $z$ is the phase difference between a signal and its delayed version. The size of the moving window was set to 2.5 cycles, with lags ranging from 1 to 3 cycles in 0.1 increments. These parameters were chosen based on *Myrov et al., 2024*. The moving window size was chosen to ensure stability of the results without excessive smoothing. The selected lag range was based on the consideration that too small lags are predominantly influenced by filtering effects, whereas large lags may capture activity from non-local sources (V. Myrov, personal communication, January 21, 2025).

To compute pACF, the signals were convolved with complex Morlet wavelets (width: three cycles). We did not use superlets here, as they were primarily designed for power estimation rather than phase analysis. In fact, when we tested superlets, the pACF values increased linearly with frequency, probably due to the way superlets combine wavelets of different widths, which can affect the phase autocorrelation of the convolved signal. To ensure an accurate comparison between FOOOF and pACF, we repeated the FOOOF decomposition using the same Morlet wavelets, which gave results comparable to those obtained with superlets, particularly at lower frequencies.

To directly compare FOOOF and pACF results, we used linear mixed models, following the same approach used for the comparison between baseline-corrected activity and FOOOF. In this analysis, pACF served as the dependent variable, while FOOOF parameters were included as fixed effect predictors, with condition modelled as a random effect. We fitted a separate model for each FOOOF parameter and estimated the marginal $R^2$.

## Comparisons between experimental conditions

We performed statistical comparisons between conditions using linear mixed models with the fitlme function in MATLAB. An advantage of using linear mixed models is their ability to accommodate missing data, which improves parameter estimates through partial pooling. We fitted the models as follows:

$$y \sim 1 + modality * nback * stimulus + rt + (1|subject) \tag{7}$$

Here, $y$ refers to the exponent or offset in the case of aperiodic activity and to the log-power in the case of periodic activity. We included reaction times ($rt$) in the model to control for differences between conditions and to assess the relationship between motor response and EEG activity.

To estimate representative reaction times for each participant and condition, we removed reaction times above a $z$-value of 3 and those below 200ms within each participant and condition. We then

fitted an ex-Gaussian distribution to estimate the parameter μ using the maximum likelihood method as implemented in the retimes package in R (**Massidda, 2013**). The ex-Gaussian distribution, a positively skewed distribution, is useful for modelling reaction time distributions (**Balota and Yap, 2011**), and the parameter μ can be interpreted as the mode of the distribution. Other parameters (e.g. $\sigma$, $\tau$) were excluded to avoid overfitting and because our aim was to obtain a simple, robust estimate of the central tendency rather than to model the entire distribution.

We fitted models in a mass univariate manner, that is, for each channel, frequency (where applicable), and time point separately.

Fixed effects were estimated using the maximum likelihood method, and the degrees of freedom were approximated using the Satterthwaite equation. We corrected the p-values using the Benjamini-Yekutieli false discovery rate (FDR) correction (**Benjamini and Yekutieli, 2001**). Given that under the false discovery rate (FDR) correction, a $q$ proportion of significant p-values are false positives, and that EEG effects occur in clusters (in time, frequency, and/or space) (**van Ede and Maris, 2016**), we additionally considered as significant only clusters of p-values that spanned either at least three time points or at least three channels. We set the significance threshold at $\alpha = .05$.

For the purposes of visualisation, p-values were averaged across channels (for heatmaps or lines) or across time (for topographies). Consequently, there are some discrepancies between heatmaps/lines and topographies.

## Comparison of aperiodic activity with event-related potentials

To rule out the possibility that aperiodic activity merely reflected evoked potentials, we first removed evoked activity before time-frequency decomposition (as described above). We then estimated associations between event-related potentials (ERPs) and aperiodic activity. Specifically, we used linear mixed models to assess this association at each channel-time point, with ERP activity as a predictor of aperiodic activity and condition included as a random effect, similar to the comparison between FOOOF parameters and baseline-corrected data.

In addition, we visually compared the topographies of aperiodic activity with those of stimulus-locked and response-locked ERPs.

The results are shown in the Supplement.

## Replication: n-back task

To test the generalisability of the results, we performed two replication analyses using two different EEG datasets: an n-back task and an item-recognition task. For the n-back task, we used an EEG dataset from a previously published study by **Nakuci et al., 2023**. This dataset includes data from 21 subjects (17 female, mean age 42±12 years) who performed a verbal n-back task (0-, 1-, and 2-back) while being recorded with a 256-channel HydroCel Geodesic Sensor Net. The task involved 150 trials per condition, 50 of which were targets. Each stimulus was presented for 400 ms, followed by an interstimulus interval of 2000 ms.

The dataset was used in its preprocessed form, as the unprocessed data was not available. The preprocessing included band-pass filtering, artifact subspace reconstruction (ASR) (**Mullen et al., 2013**), independent component analysis, channel interpolation, another round of ASR, re-referencing to the average reference, epoching from –1000–2000 ms around stimulus presentation, and baseline correction using the prestimulus period (**Nakuci et al., 2023**). Notably, the data were band-pass filtered between 0.5 and 50 Hz, so we restricted the time-frequency decomposition and FOOOF analysis to frequencies between 3 and 35 Hz.

We carried out all other analytical steps, including time-frequency decomposition, FOOOF analysis, and statistical comparisons, in the same manner as in the primary analysis.

The results of these analyses are presented in the Supplement.

## Replication: item-recognition task

To test generalisability across tasks, we analysed EEG recordings from another study, which included 48 participants (10 male, mean age 23.8±5 years). All participants were right-handed, had normal or corrected-to-normal vision, and were free of neurological or psychiatric illness. Most participants were psychology students at the University of Ljubljana, while the rest responded to invitations sent via email

and social media. All participants provided informed consent, approved by the Ethics Committee of the Faculty of Arts, University of Ljubljana.

In this study, we used a Sternberg item-recognition task. In this paradigm, a series of visual target stimuli is presented, followed by a probe stimulus that may or may not be identical to one of the previous targets. Participants' task is to indicate whether the probe (test) stimulus was part of the initial set (*Sternberg, 1969*).

The experiment included three task conditions performed under two working memory loads: (1) presentation of two or four stimuli that differed in colour but were displayed in the same location; (2) presentation of two or four stimuli that differed in position but were of the same colour (black); (3) presentation of two or four stimuli that differed in both colour and position (integration). Each condition consisted of 432 trials, with 216 trials per working memory load, divided into 18 blocks of 48 trials each. Participants were randomly assigned to three groups, each performing the task in two of the three conditions, resulting in 32 participants per condition and 16 participants per comparison of conditions. A schematic diagram and a detailed description of the task can be found in *Figure 10—figure supplement 1*.

Stimuli were presented on a 27-inch LCD screen with a refresh rate of 120 Hz. The stimuli were presented on a grey background in the centre of the screen, within a frame 250 pixels wide and 250 pixels high. The stimuli appeared as circles 20 pixels in diameter. When different positions were required, the stimuli were spaced at a distance equal to six times the stimulus radius. The frame spanned approximately 4.05 degrees of vertical and horizontal visual angle, and the diameter of each stimulus spanned approximately 0.32 degrees of visual angle. The task was programmed and ran using PsychoPy 2 (*Peirce et al., 2022*).

For the EEG measurements, we used a BrainAmp EEG amplifier (Brain Products GmbH, Germany) and a 32-channel ActiCAP EEG cap, configured according to the international 10/20 system. Participants provided responses using a Cedrus response pad, RB series, model 530. We monitored horizontal eye movements using a camera and a dedicated HEOG channel ([F7—F8]/2). We manually removed trials in which participants tracked the target stimuli with their eyes.

We preprocessed the EEG data using the same pipeline as for the n-back task, except that the data were segmented into epochs ranging from −1700–5400 milliseconds, time-locked to the cue presentations. Subsequent EEG analyses were performed in a manner identical to the n-back task, including only epochs with correct responses.

## Acknowledgements

The authors would like to thank Zvezdan Pirtošek for the help with project management, Katarina Marjanovič, Aleš Oblak, Katarina Sinja Miloševič, Tala Koren, Anja Čuš, Nina Lang, Taja Žnidarič, Dunja Kolenko, Irena Hostnikar, Urška Levac, Matija Kuclar, Vesna Muzek, Klara Babič, Kristina Klančič, Sara Bugarinovič, and Lea Kukovec for the help with the data collection. The authors would also like to thank the participants, without whom this study would not have been possible.

## Additional information

### Competing interests

Andraž Matkovič: Has previously consulted for Neumora. Grega Repovš: Consults for and holds equity in Neumora and Manifest Technologies. The other authors declare that no competing interests exist.

### Funding

| Funder | Grant reference number | Author |
| --- | --- | --- |
| The Slovenian Research and Innovation Agency | J3-8200 | Tisa Frelih<br>Andraž Matkovič<br>Tjaša Mlinarič<br>Jurij Bon |
| The Slovenian Research and Innovation Agency | P5-0110 | Andraž Matkovič<br>Jurij Bon |

| Funder | Grant reference number | Author |
|---|---|---|
| The Slovenian Research and Innovation Agency | P3-0338 | Grega Repovš |
| The Slovenian Research and Innovation Agency | J5-4590 | Andraž Matkovič Jurij Bon Grega Repovš |
| The Slovenian Research and Innovation Agency | J3-9264 | Andraž Matkovič Tjaša Mlinarič Grega Repovš |

The funders had no role in study design, data collection and interpretation, or the decision to submit the work for publication.

## Author contributions

Tisa Frelih, Conceptualization, Investigation, Visualization, Methodology, Writing – original draft, Project administration, Writing – review and editing; Andraž Matkovič, Conceptualization, Data curation, Software, Formal analysis, Investigation, Visualization, Methodology, Writing – original draft, Writing – review and editing; Tjaša Mlinarič, Investigation, Project administration, Writing – review and editing; Jurij Bon, Resources, Supervision, Writing – review and editing; Grega Repovš, Conceptualization, Resources, Software, Supervision, Methodology, Writing – review and editing

## Author ORCIDs

Tisa Frelih (ID) https://orcid.org/0009-0008-8671-0922
Andraž Matkovič (ID) https://orcid.org/0000-0003-3829-3040

## Ethics

Human subjects: The study protocol was approved by the Medical Ethics Committee of the Republic of Slovenia (protocol number 0120-128/2019/9; 21.08.2019). All study participants gave written informed consent prior to the initiation of any study-related procedures.

Reviewer #1 (Public review): https://doi.org/10.7554/eLife.101071.4.sa1
Reviewer #2 (Public review): https://doi.org/10.7554/eLife.101071.4.sa2
Reviewer #3 (Public review): https://doi.org/10.7554/eLife.101071.4.sa3
Author response https://doi.org/10.7554/eLife.101071.4.sa4

---

# Additional files

## Supplementary files

MDAR checklist

## Data availability

The preprocessed data are available at https://dx.doi.org/10.17605/OSF.IO/RQ8ZM. The n-back dataset from *Nakuci et al., 2023* is available at https://zenodo.org/records/6897260. Preprocessing was performed using EEGLAB (https://github.com/sccn/eeglab; *Delorme et al., 2025*) and EEGLAB plugins. Time-frequency decomposition was performed using the Fieldtrip function ft_freqanalysis (https://github.com/fieldtrip/fieldtrip/blob/master/ft_freqanalysis.m; *Oostenveld et al., 2025*). FOOOF decomposition was performed using the Python package specparam (https://github.com/fooof-tools/fooof/; *Donoghue, 2025*). Code for phase-autocorrelation function is available on GitHub: https://github.com/palvalab/discovering_rhythmicity; *Vlad0922, 2024*. Linear mixed models were estimated using the MATLAB function fitlme (https://www.mathworks.com/help/stats/fitlme.html). Visualisation was done using the packages ggplot2 (https://ggplot2.tidyverse.org/), matplotlib (https://matplotlib.org/) and MNE-Python (https://mne.tools/).

The following dataset was generated:

| Author(s) | Year | Dataset title | Dataset URL | Database and Identifier |
|---|---|---|---|---|
| Frelih T, Matkovič A, Mlinarič T, Bon J, Repovš G | 2024 | Modulation of aperiodic EEG activity provides sensitive index of cognitive state changes during working memory task | https://doi.org/10.17605/OSF.IO/RQ8ZM | Open Science Framework, 10.17605/OSF.IO/RQ8ZM |

The following previously published dataset was used:

| Author(s) | Year | Dataset title | Dataset URL | Database and Identifier |
|---|---|---|---|---|
| Nakuci J, Covey T, Shucard T, Shucard D, Muldoon S | 2022 | Single trial variability in neural activity during a working memory task reveals multiple distinct information processing sequences | https://doi.org/10.5281/zenodo.6897260 | Zenodo, 10.5281/zenodo.6897260 |

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

# Appendix 1

## Simulation

To assess whether the FOOOF decomposition can reliably separate low-frequency periodic activity from the aperiodic background, we ran a simulation in which the periodic parameters were systematically varied. Specifically, we used the sim_spectrum function from the Python package specparam (https://github.com/fooof-tools/fooof, copy archived at *Donoghue, 2025*) to generate 100 power spectrum instances in the range 1–30 Hz, with a frequency resolution of 1 Hz to match our experimental analyses. The offset was set to 1 and the exponent to 1.5. A single periodic component was added to each instance using the following parameter variations:

- central frequency: 3, 4, 5, 6, 7 Hz,
- periodic power: 0.2, 0.4, 0.8, 1.6,
- bandwidth: 0.5, 1, 2, 4,

Gaussian noise with a mean of 0 and a standard deviation of 0.1 was added to each power spectrum. Fitting was performed using the 'fixed' method, estimating only the aperiodic offset and exponent parameters (without the knee parameter). We performed two fitting procedures: first, across the entire frequency range (1–30 Hz) (panel A in the figures below), and second, within a limited range of 3–30 Hz (panel B), consistent with our experimental EEG data analysis, where FOOOF parameters were estimated only for frequencies ≥3 Hz. Simulated data are shown in *Appendix 1—figure 1*.

Estimates of the exponent and offset parameters were generally accurate at low bandwidths and low periodic power levels (*Appendix 1—figures 2 and 3*). However, fitting within the restricted range (≥3 Hz, panel B in all figures) resulted in reduced accuracy, especially when the periodic component had a lower central frequency. In cases where the periodic component had large bandwidths and/or low power, it was often conflated with the aperiodic component, inflating the aperiodic parameter estimates. This effect was particularly pronounced when only frequencies above 3 Hz were fitted, as indicated by a high number of unestimable periodic components (*Appendix 1—figures 4–6*), especially when the simulated central frequency was low (i.e. 3 Hz). Inflated aperiodic parameter estimates were associated with increased central frequency estimates and decreased periodic power estimates. Central frequency and bandwidth estimates showed high variability at low power levels. Conversely, periodic power estimates were more accurate at low simulated power, but less so when both bandwidth and power were high.

Notably, the model fit indices ($R^2$ and model error) were generally good, even in cases where periodic components were poorly estimated or not detectable at all (*Appendix 1—figure 7 and 8*). This suggests that model fit metrics can be misleading – high fit does not necessarily imply correct parameter estimation.

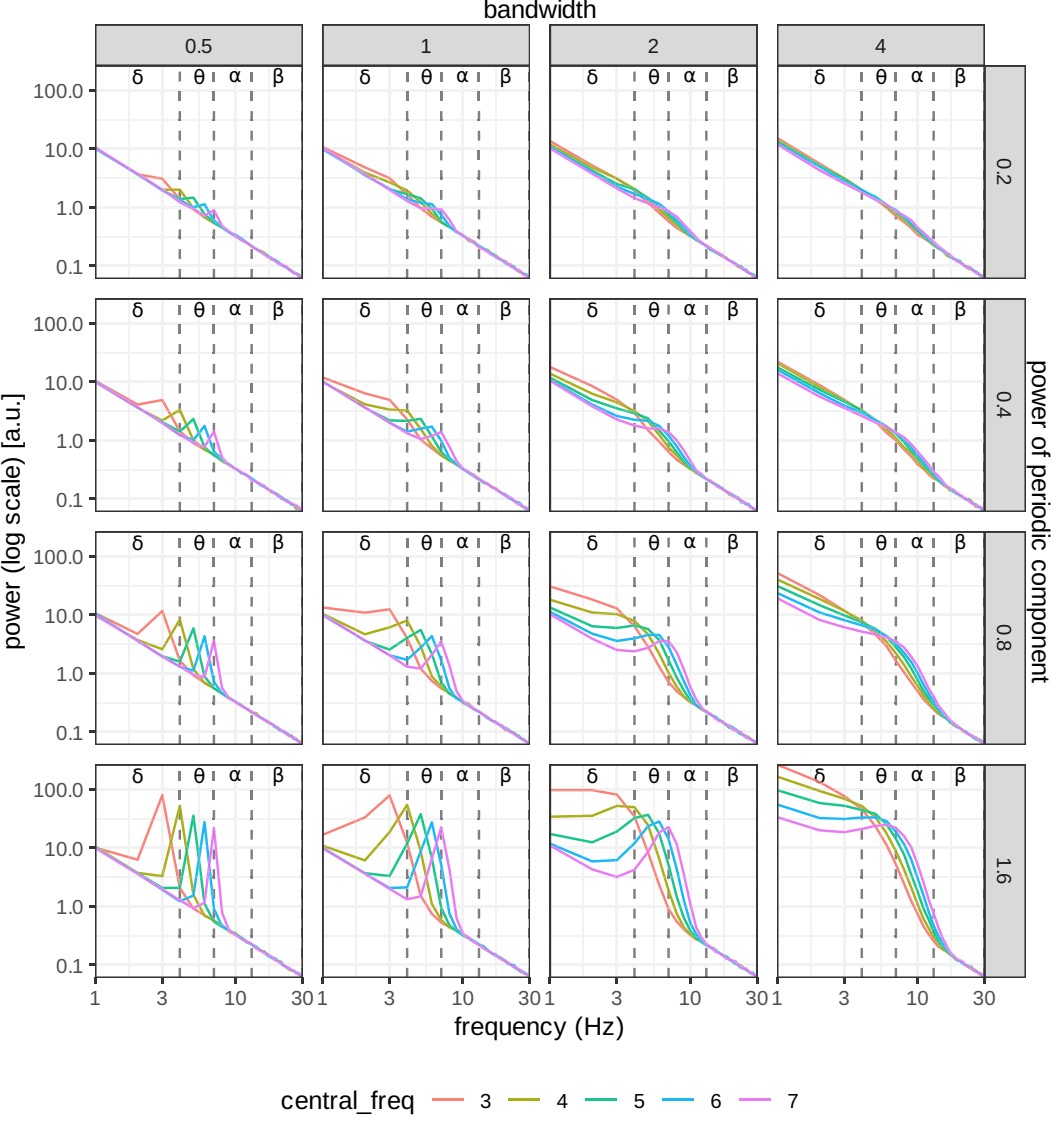

**Appendix 1—figure 1.** Simulated power spectra in log-log space. We added a single periodic component to each power spectrum, with the exponent fixed at 1.5 and the offset at 1.

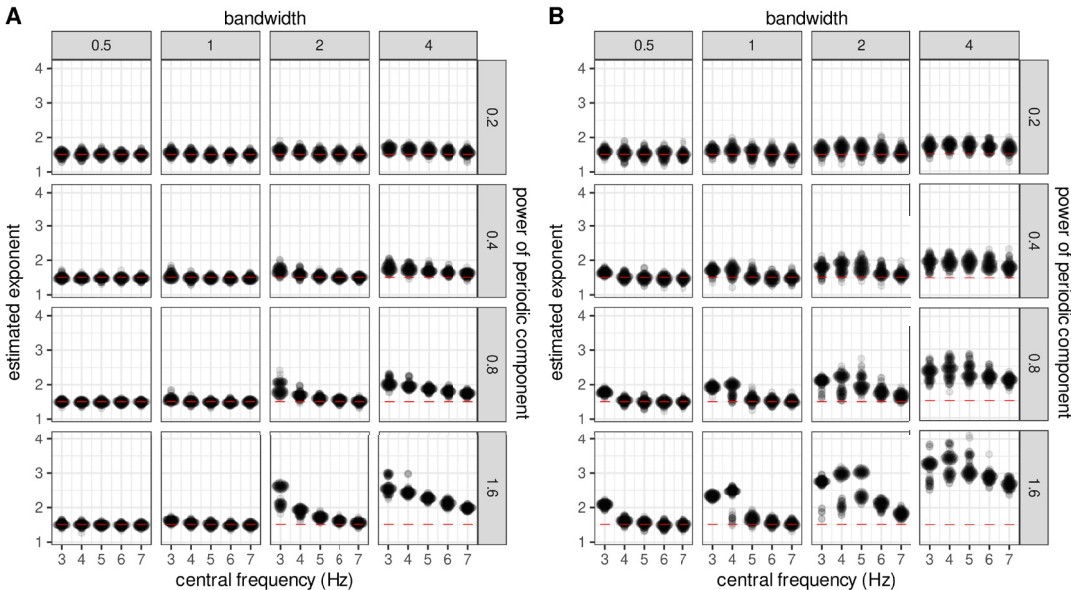

**Appendix 1—figure 2.** Exponent estimates. Red lines indicate ground truth. Panel A covers 1–30 Hz fits, while panel B covers 3–30 Hz fits. Exponent estimates were inflated at larger bandwidths and higher periodic power, especially when the periodic component's central frequency was low. This suggests a mixing of periodic and aperiodic components.

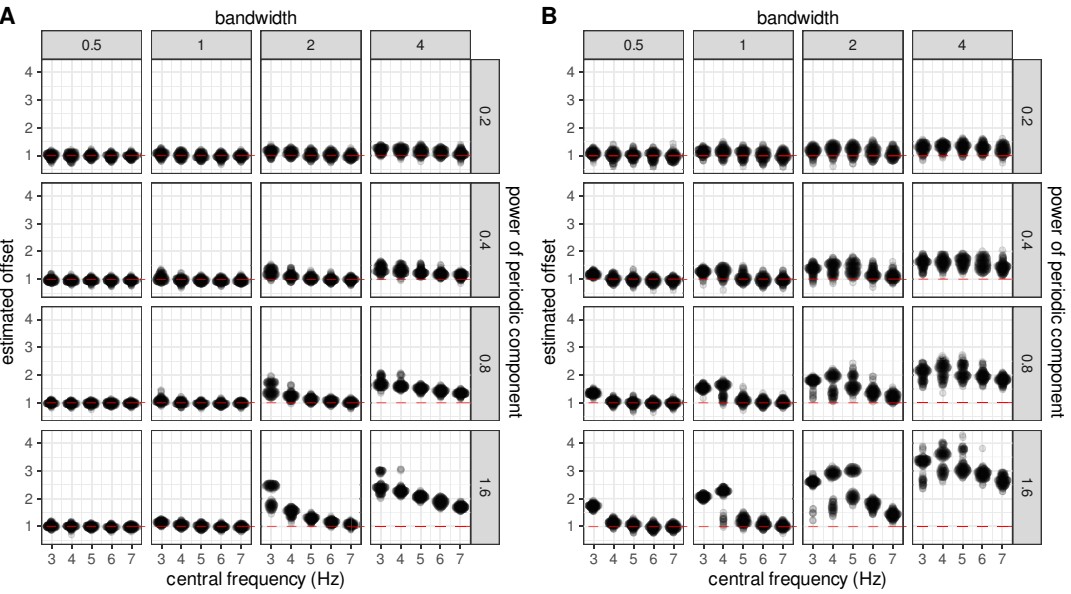

**Appendix 1—figure 3.** Offset estimates. As in *Appendix 1—figure 2*, estimates tended to be inflated when the periodic component was difficult to detect.

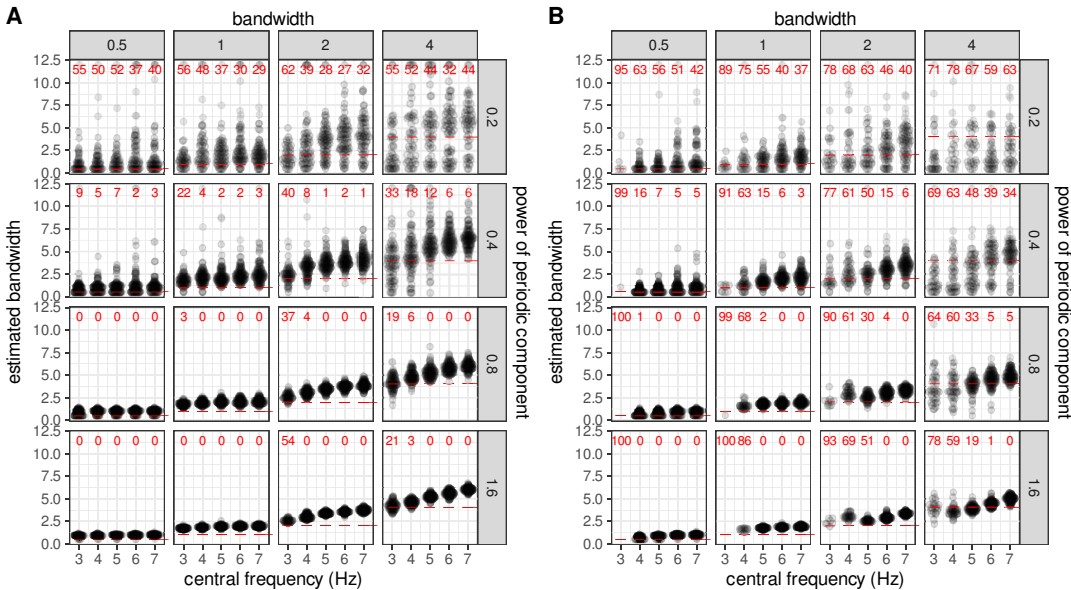

**Appendix 1—figure 4.** Bandwidth estimates. Red lines show ground truth, while red numbers mark how often the periodic component was undetectable (out of 100 cases). Low frequency, low power, and the restricted 3–30 Hz fitting range often made periodic features hard to detect.

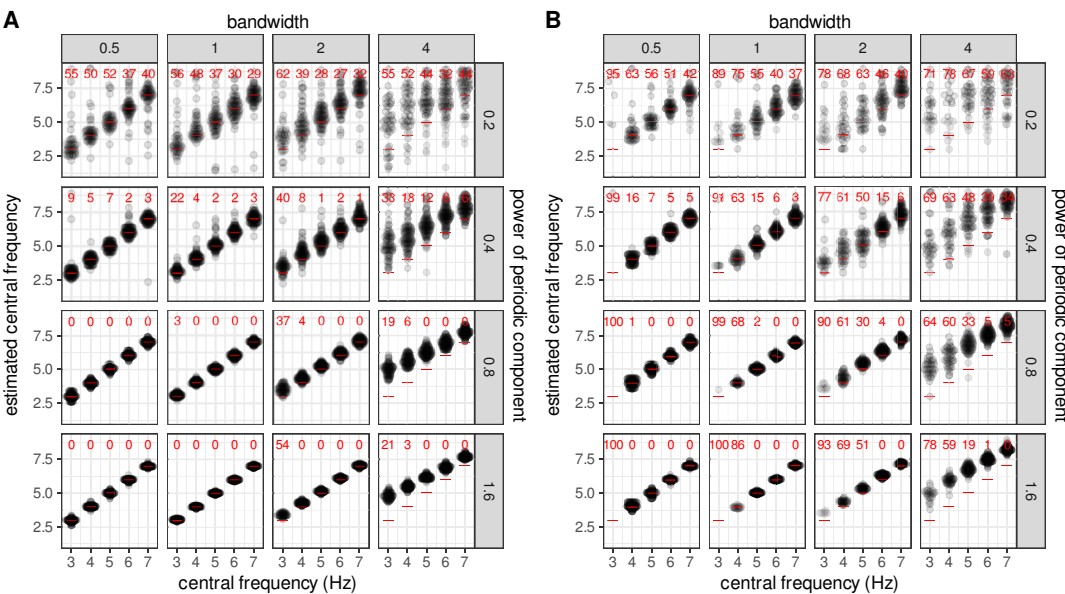

**Appendix 1—figure 5.** Central frequency estimates. Central frequency estimates were inflated for large bandwidths and high power, particularly when fitting was restricted to ≥3 Hz (panel B). Variability of estimates also increased at low simulated power. The red numbers indicate how many times out of 100 the periodic component was undetectable.

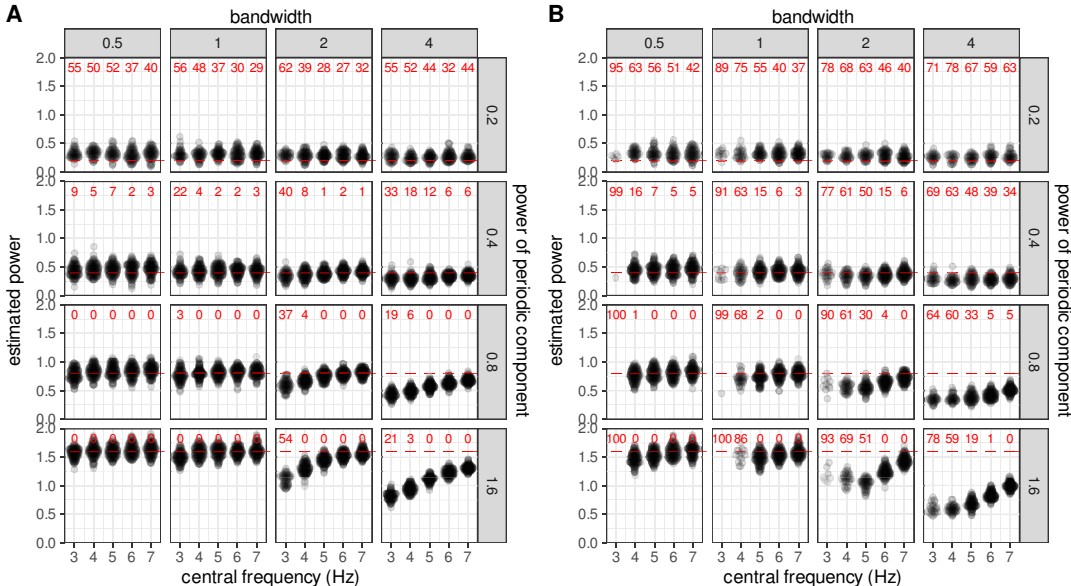

**Appendix 1—figure 6.** Power estimates. In contrast to bandwidth and central frequency estimates, power estimates remained largely accurate at low simulated power, regardless of bandwidth. However, at high simulated power and large bandwidths, power estimates were deflated. The red numbers indicate how many times out of 100 the periodic component was undetectable.

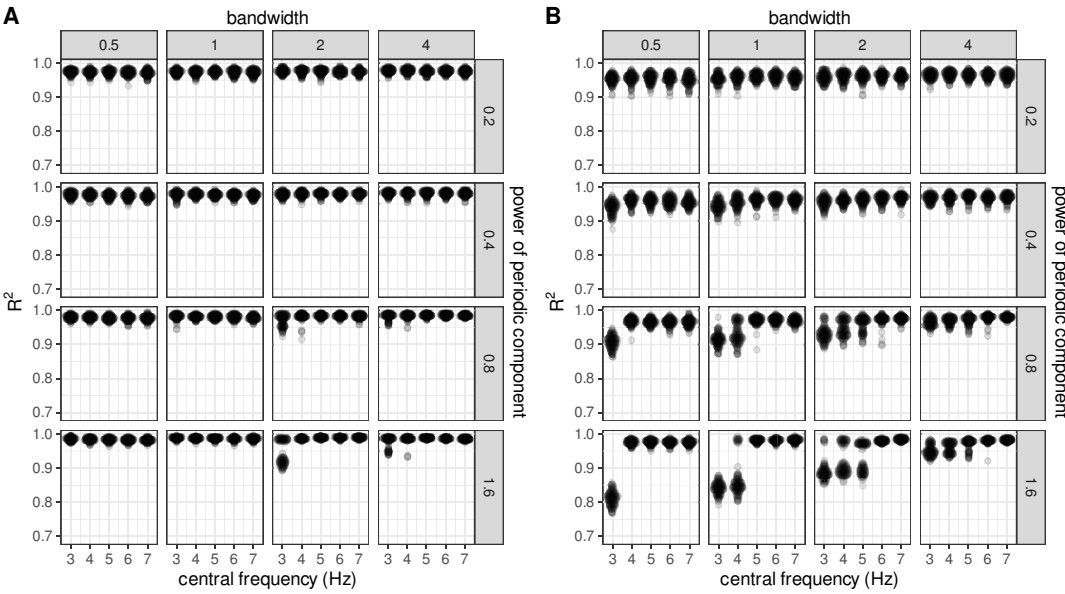

**Appendix 1—figure 7.** $R^2$ index of model fit. $R^2$ values were very high in most cases, even when parameter estimates were largely inaccurate (see previous figures). The lowest $R^2$ values were observed for low bandwidths, low central frequencies, and high power.

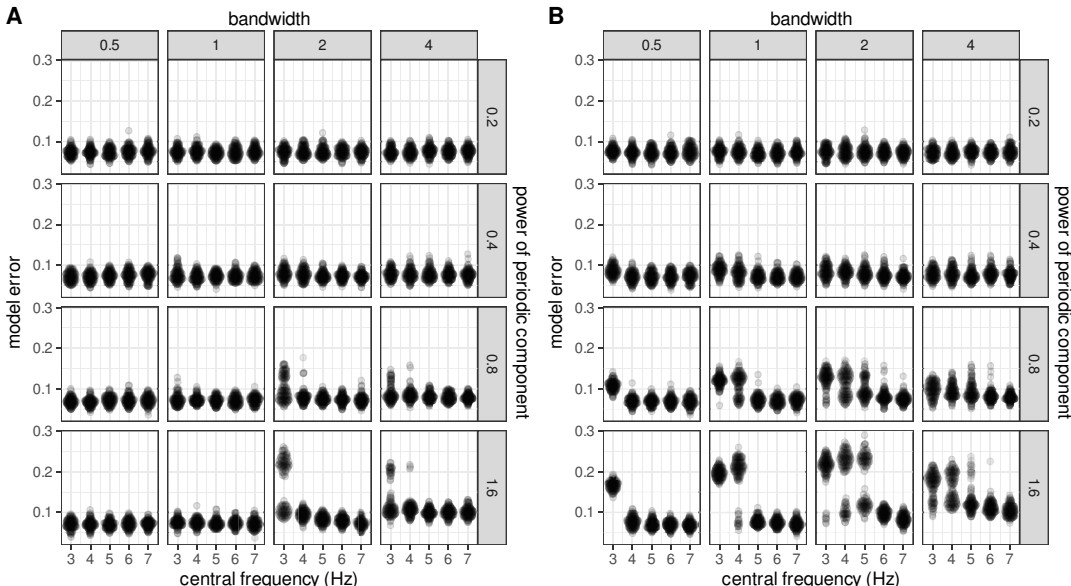

**Appendix 1—figure 8.** Model errors. Similar to $R^2$, higher model errors were associated with low central frequencies and large powers, and the effect was more pronounced when only frequencies ≥3 Hz were fitted.

## Appendix 2

### Goodness-of-fit measures

Here, we report the $R^2$ values and model error metrics used to assess the goodness-of-fit of the FOOOF models. However, as demonstrated in the simulation (Appendix 1), such fit metrics can sometimes be misleading.

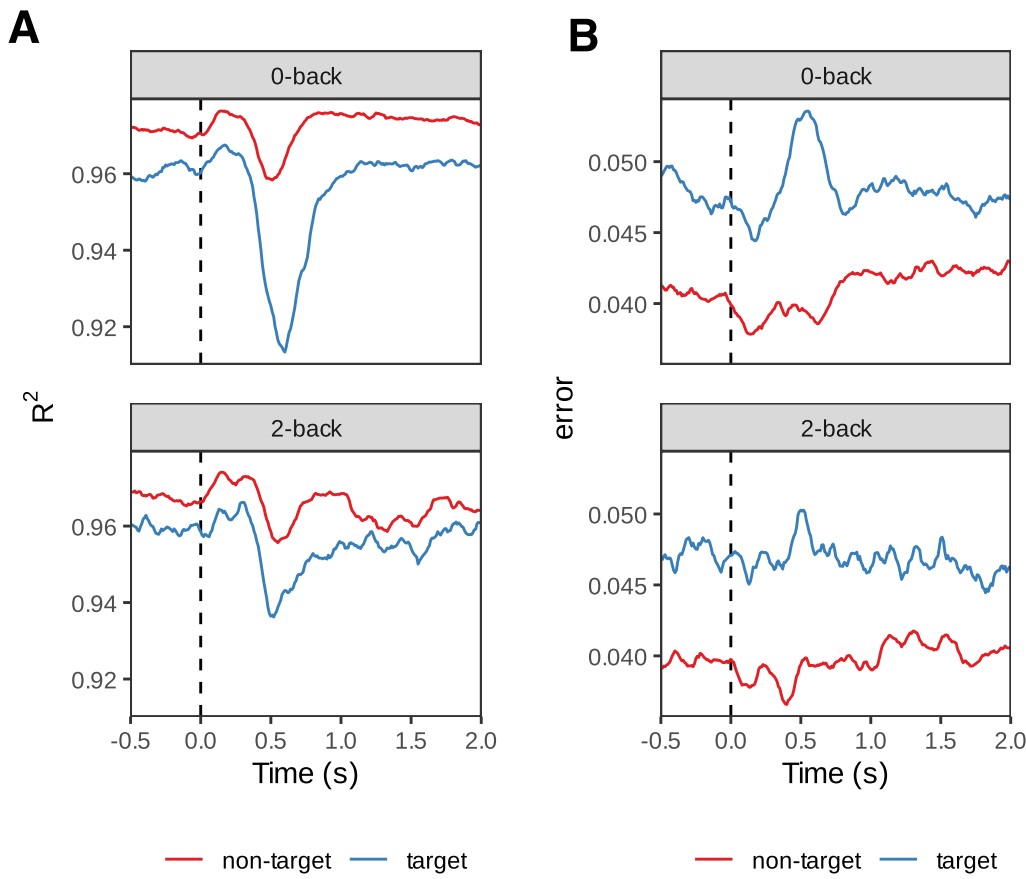

**Appendix 2—figure 1.** Fitting Oscillations and One-Over-F (FOOOF) goodness-of-fit measures. Averaged across channels, subjects, and conditions. Goodness-of-fit was lowest around 0.5 s after stimulus onset, corresponding to the decrease in aperiodic and periodic activity.

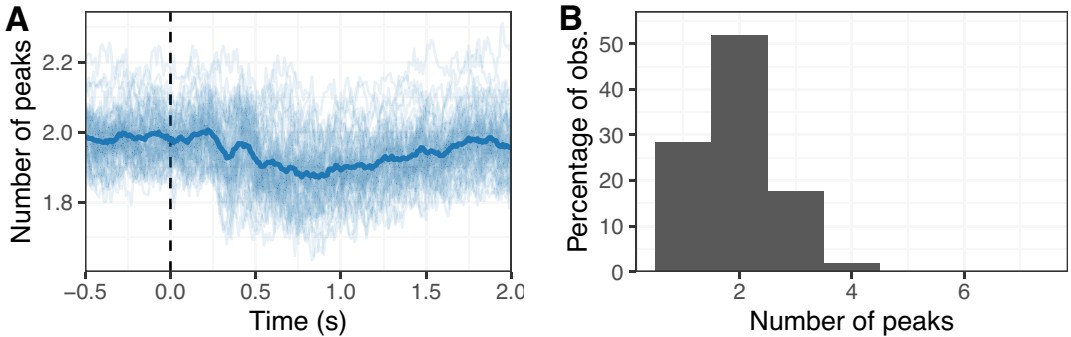

**Appendix 2—figure 2.** Number of identified peaks in FOOOF models. (**A**) Average number of peaks across all subjects, channels, and conditions. Light lines represent individual channels. (**B**) Histogram of the number of peaks
*Appendix 2—figure 2 continued on next page*

*Appendix 2—figure 2 continued*
in all models. On average, 1.9–2 peaks were identified per model, showing high consistency across models and indicating that the models did not overfit by detecting an excessive number of peaks.

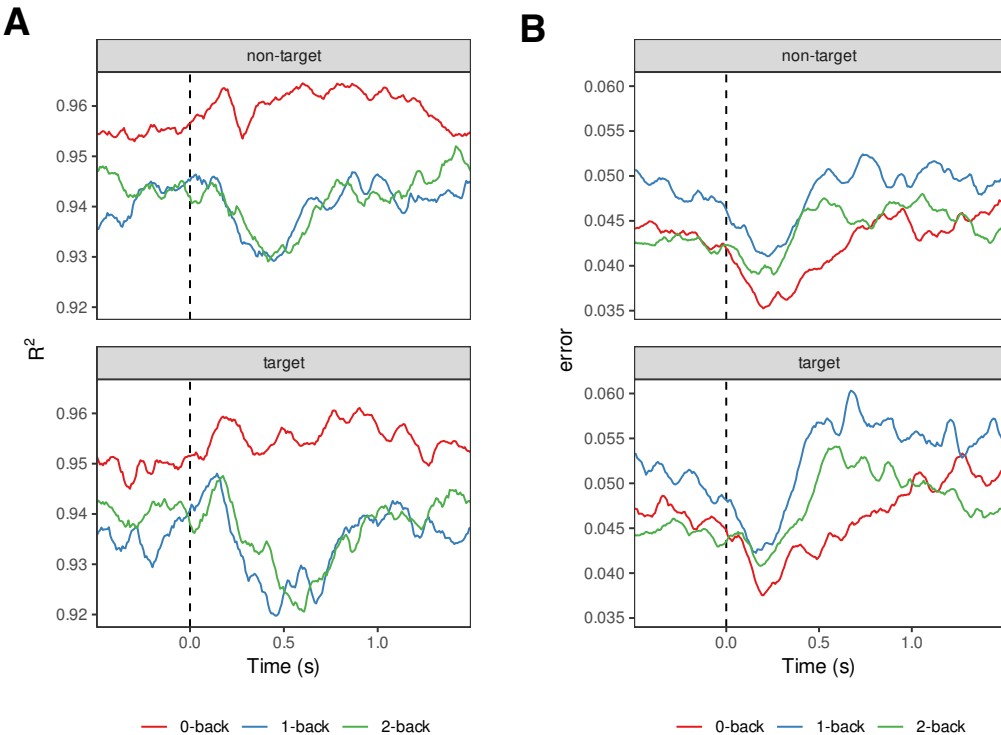

**Appendix 2—figure 3.** FOOOF goodness-of-fit measures (control dataset). Averaged across channels, subjects and conditions. Similar to our primary dataset (*Appendix 2—figure 1*), the decrease in goodness-of-fit around 0.5 s after stimulus onset coincided with the decrease in periodic activity.

**A**

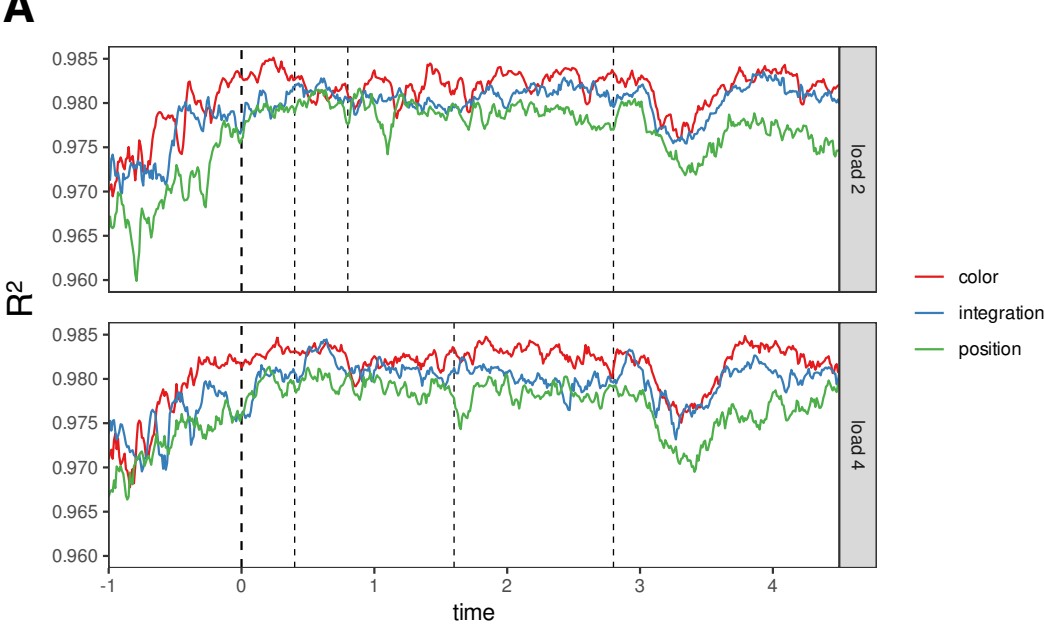

**B**

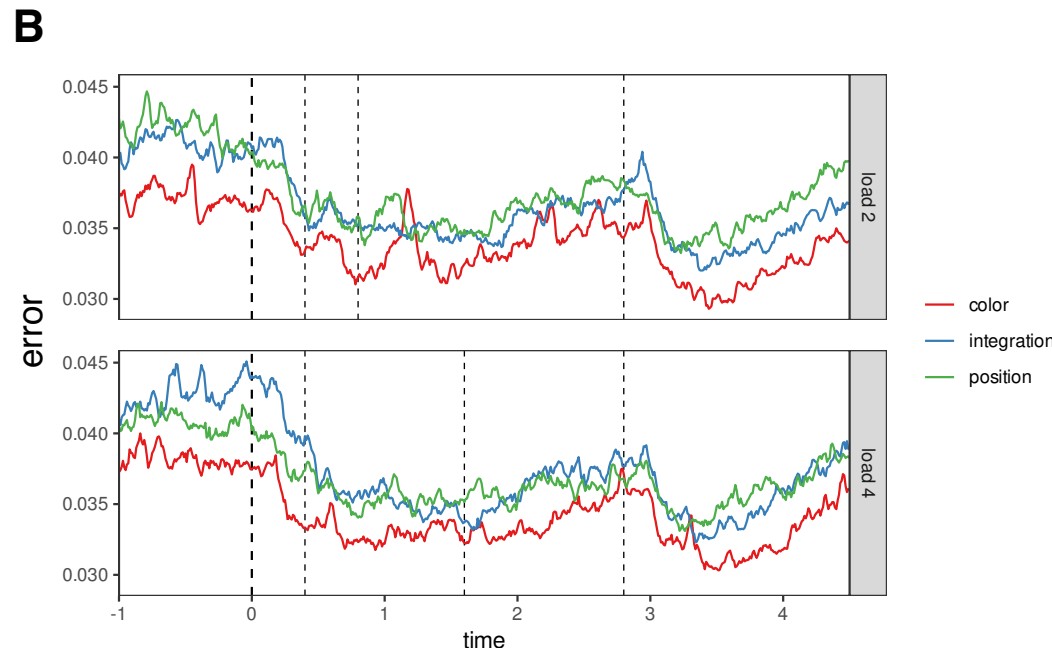

**Appendix 2—figure 4.** FOOOF goodness-of-fit measures for the item-recognition task, averaged across channels. $R^2$ was above 0.97 and model error was below 0.045 throughout the task. There was a small decrease in goodness-of-fit around 3 s after stimulus onset, coinciding with a decrease in periodic activity, similar to the other two datasets (*Appendix 2—figures 1 and 3*).

