## [Editor Report · eLife Assessment]

This **valuable** work explores the timely idea that aperiodic activity in human electrophysiology recordings is dynamically modulated in response to task events in a manner that may be relevant for behavioral performance. Moreover, the authors present **solid** evidence that, in some circumstances, these aperiodic changes might be misinterpreted as oscillatory changes.

---

## [Referee Report · Reviewer #1 (Public review)]

Summary:

Frelih et al. investigated both periodic and aperiodic activity in EEG during working memory tasks. In terms of periodic activity, they found post-stimulus decreases in alpha and beta activity, while in terms of aperiodic activity, they found a bi-phasic post-stimulus steepening of the power spectrum, which was weakly predictive of performance. They conclude that it is crucial to properly distinguish between aperiodic and periodic activity in event-related designs as the former could confound the latter. They also add to the growing body of research highlighting the functional relevance of aperiodic activity in the brain.

Strengths:

This is a well-written, timely paper that could be of interest to the field of cognitive neuroscience, especially to researchers investigating the functional role of aperiodic activity. The authors describe a well-designed study that looked at both the oscillatory and non-oscillatory aspects of brain activity during a working memory task. The analytic approach is appropriate, as a state-of-the-art toolbox is used to separate these two types of activity. The results support the basic claim of the paper that it is crucial to properly distinguish between aperiodic and periodic activity in event-related designs as the former could confound the latter. They also add to the growing body of research highlighting the functional relevance of aperiodic activity in the brain. Commendably, the authors include replications of their key findings on multiple independent data sets.

Comments on the previous version:

The authors have addressed several of the weaknesses I noted in my original review, specifically, they softened their claims regarding the theta findings, while simultaneously strengthening these findings with additional analyses (using simulations as well as a new measure of rhythmicity, the phase autocorrelation function, pACF). Most of the other suggested control analyses were also implemented. While I believe the fact that the participants in the main sample were not young adults could be made even more explicit, and the potential interaction between age and aperiodic changes could be unpacked a little in the discussion, the age of the sample is definitely addressed upfront.

---

## [Referee Report · Reviewer #2 (Public review)]

Summary:

In this manuscript, Frelih et al, investigate the relationship between aperiodic neural activity, as measured by EEG, and working memory performance, and compares this to the more commonly analyzed periodic, and in particular theta, measures that are often associated with such tasks. To do so, they analyze a primary dataset of 57 participants engaging in an n-back task, as well as a replication dataset, and use spectral parameterization to measure periodic and aperiodic features of the data, across time. In the revision, the authors have clarified some key points, and added a series of additional analyses and controls, including the use of an additional method, that helps to complement the original analyses and further corroborates their claims. In doing so, they find both periodic and aperiodic features that relate to the task dynamics, but importantly, the aperiodic component appears to explain away what otherwise looks like theta activity in a more traditional analysis. This study therefore helps to establish that aperiodic activity is a task-relevant dynamic feature in working memory tasks and may be the underlying change in many other studies that reported 'theta' changes, but did not use methods that could differentiate periodic and aperiodic features.

Strengths:

Key strengths of this paper include that it addresses an important question - that of properly adjudicating which features of EEG recordings relate to working memory tasks - and in doing so provides a compelling answer, with important implications for considering prior work and contributing to understanding the neural underpinnings of working memory. The revision is improved by showing this using an additional analysis method. I do not find any significant faults or error with the design, analysis, and main interpretations as presented by this paper, and as such, find the approach taken to be a valid and well-enacted. The use of multiple variants of the working memory task, as well as a replication dataset significantly strengthens this manuscript, by demonstrating a degree of replicability and generalizability. This manuscript is also an important contribution to motivating best practices for analyzing neuro-electrophysiological data, including in relation to using baselining procedures. I think the updates in the revision have helped to clarify the findings and impact of this study.

Weaknesses:

Overall, I do not find any obvious weaknesses with this manuscript and it's analyses that challenge the key results and conclusions. Updates through the revision have addressed my previous points about adding some additional notes on the methods and conclusions.

---

## [Referee Report · Reviewer #3 (Public review)]

Summary:

Using a specparam (1/f) analysis of task-evoked activity, the authors propose that "substantial changes traditionally attributed to theta oscillations in working memory tasks are, in fact, due to shifts in the spectral slope of aperiodic activity." This is a very bold and ambitious statement, and the field of event-related EEG would benefit from more critical assessments of the role of aperiodic changes during task events. Unfortunately, the data shown here does not support the main conclusion advanced by the authors.

Strengths:

The field of event-related EEG would benefit from more critical assessments of the role of aperiodic changes during task events. The authors perform a number of additional control analyses, including different types of baseline correction, ERP subtraction, as well as replication of the experiment with two additional datasets.

Comments on previous revisions:

The authors have completed a substantial revision based on the comments from all of the reviewers. Overall, the major claims of the initial report have been profoundly tempered.

[Editors' note: We determined that this revised version appropriately tempers some of the prior claims and addresses the concerns raised by the reviewers through two rounds of review.]

---

## [Author Response]

The following is the authors’ response to the previous reviews

**Reviewer 1:**

We thank Reviewer 1 for the discussion on the possible causes of ERPs and their relevance for the interpretation of changes in aperiodic activity. We have changed the relevant paragraph to read as follows: For example, ERPs may reflect changes in periodic activity, such as phase resets (Makeig et al., 2002), or baseline shifts (Nikulin et al., 2007). ERPs may also capture aperiodic activity, either in the form of evoked transients triggered by an event (Shah et al., 2004) or induced changes in the ongoing background signal. This has important implications: evoked transients can alter the broadband spectrum without implying shifts in ongoing background activity, whereas induced aperiodic changes may signal different neural mechanisms, such as shifts in the excitation-inhibition balance (Gao et al., 2017).

Reviewer 1 argued that a time point-by-time point comparison between ERPs and aperiodic parameters may not be the most appropriate approach, since aperiodic time series have lower temporal resolution than ERPs. Reviewer suggested comparing their topographies instead. We had already done this in the first version of the paper (see Fig. S7: https://elifesciences.org/reviewedpreprints/101071v1#s10). However, in the second version, we opted to use linear mixed models for each channel-time point in order to maintain consistency with the other analyses in the paper (e.g. the comparison between FOOOF parameters and baseline-corrected power).

Nevertheless, we repeated the topographic correlations as in the first version, and the results are shown below. Correlations were computed for each time point, subject and condition, and then averaged across these dimensions for visualisation. The pattern differs from that of the linear mixedmodel results (see Fig. S14), with notable correlations appearing after ~0.5 s for the exponent and after ~1.0 s for the offset. Still, the correlations remain low, suggesting that aperiodic parameters and ERPs encode different information (at least in this dataset).

**Author response image 1. sa4fig1:** 

Additionally, to control for the effect of smearing we have performed the same linear mixed model analysis as in Fig. S14 on low-pass filtered ERPs (with cut-off 10 Hz), and the results were largely similar as in Fig. S14.

Reviewer 1 discussed two possible explanations for the observed correlations between baselinecorrected power and FOOOF parameters (Figure 4): “The correlation between the exponent and lowfrequency activity could be of either direction: low frequency power changes could reflect 1/f shifts, or exponent estimates might be biased by undetected delta/theta activity. I think that one other piece of evidence /…/ to intuitively highlight why the latter is more likely is the /…/ decrease at high ("transbeta") frequencies, which suggests a rotational shift /../.” We agree with the interpretation that lowfrequency power changes in our data primarily reflect 1/f shifts. However, we are uncertain about the reviewer’s statement that the “latter” explanation (i.e., bias in exponent estimates due to delta/theta activity) is more likely. Given the context, we believe the reviewer may have intended to say the “former” explanation is more likely.

We agree with the reviewers' observation that rhythmicity, as estimated using the pACF, can be independent of power (Myrov et al., 2024, Fig. 1). However, it seems that in real (non-simulated) datasets, the pACF and power spectral density (PSD) are often moderately correlated (e.g. Myrov et al., 2024, Fig. 5).

Reviewer 1 asked whether we had examined aperiodic changes in the data before and after subtracting the response-locked ERPs. We did not carry out this extra analysis as, as the reviewer suggests, it would have been excessive – the current version of the paper already contains more than 60 figures. As mentioned in the manuscript, we acknowledge the possibility that response-locked ERPs contribute to the second aperiodic component. However, due to the weak correlation between reaction times and aperiodic activity, the presence of both components throughout the entire epoch (in at least the first and third datasets) and the distinct differences between the ERPs and the aperiodic activity in the different conditions (see Fig. 8 vs. Fig. S13), we cannot conclusively determine whether the second aperiodic component is directly related to motor responses. Finally, we agree with the reviewer that the distribution of the response-locked ERP more closely resembles the frontocentral (earlier) aperiodic component than the later post-response component. We have amended the relevant paragraph in the Discussion to include these observations. ”While it is possible that response-related ERPs contributed to the second aperiodic component, several observations suggest otherwise: both aperiodic components were present throughout the entire epoch, differences between conditions diverged between ERPs and aperiodic activity (compare Figure 8 and Figure S16), and the associations with reaction times were weak. Moreover, the distribution of the response-locked ERP qualitatively resembled the earlier frontocentral aperiodic component more than the later post-response component. Taken together, these findings suggest that ERPs and aperiodic activity capture distinct aspects of neural processing, rather than reflecting the same underlying phenomenon.”

We agree with Reviewer 1 that our introduction of aperiodic activity was abrupt, and that the term 'aperiodic exponent' required definition. We have now defined it as the spectral steepness in log–log space (i.e. the slope), and have added a brief explanatory sentence to the introduction.

Reviewer 1 noted that the phrase 'task-related changes in overall power' could be misinterpreted as referring to total (broadband) power, and recommended that we specify a frequency range. We agree, so we have replaced 'overall power' with 'spectral power within a defined frequency range'.

We agree with Reviewer 1 that the way we worded things in the Discussion section regarding alpha activity and inhibitory processes was awkward and could easily be misread. We have rephrased the sentences and added a brief explanation to avoid implying a direct link between alpha attenuation and neural inhibition.

Furthermore, based on the reviewer’s suggestion, we added a brief comment in the Discussion section (Theoretical and methodological implications) on theoretical perspectives regarding the interaction between age and aperiodic activity.

Reviewer 1 suggested including condition as a fixed effect in order to examine whether the relationship between FOOOF parameters and baseline-corrected power is modulated by condition. Specifically, the reviewer proposed changing our model from

baseline_corrected_power ~ 1 + fooof_parameter + (1|modality) + (1|nback) + (1|stimulus) + (1|subject)

to

baseline_corrected_power ~ 1 + fooof_parameter + modality*nback *stimulus + (1|subject)

While we appreciate this suggestion, we believe that including design variables as fixed effects would confound the interpretation of (marginal) R² as a measure of the association between FOOOF parameters and baseline-corrected power. Our primary question in this analysis was about the fundamental relationship between these measures, not how experimental conditions moderate this relationship.

To address the reviewer's concern regarding condition-specific effects, we conducted separate analyses for each condition using a simpler model:

baseline_corrected_power ~ 1 + fooof_parameter + (1|subject)

The results (now included in the Supplement, Fig. S4–S6) show generally smaller effect sizes compared to our original random-effects model, with notable differences between conditions. The 2-back conditions, particularly the non-target trials, exhibited the weakest associations. Despite these differences, the overall patterns remained consistent with our original findings: exponent and offset exhibited positive associations at low frequencies (delta, theta) and negative associations at higher frequencies (beta, low gamma), while periodic activity correlated substantially with baselinecorrected power in the alpha, beta, and gamma ranges.

However, this condition-specific approach has important limitations. With only 47 subjects per condition, the statistical power is insufficient for stable correlation estimates (Schönbrodt & Perugini, 2013; https://doi.org/10.1016/j.jrp.2013.05.009). This likely explains why the effects are smaller and less stable effects than in our original model, which uses the full dataset's power while appropriately accounting for condition-related variance through random effects. Since these additional analyses do not alter our primary conclusions, we have included them in the Supplement for completeness and made a minor change in the Discussion section.

Reviewer 1 asked what channels are lines on Figure 9 based on. As stated in the Methods section, “We fitted models in a mass univariate manner, that is for each channel, frequency (where applicable), and time point separately. /…/ For the purposes of visualisation, p-values were averaged across channels (for heatmaps or lines) or across time (for topographies).” Therefore, the lines and heatmaps apply to all channels.

**Reviewer 2:**

We would like to thank reviewer 2 for their detailed explanation of the expected behaviour of the specparam algorithm. We have added the following explanation to the Methods section:

Importantly, as noted by the reviewer, this behaviour reflects an explicit design choice of the algorithm: to avoid overfitting ambiguous peaks at the edges of the spectrum, FOOOF excludes peaks that are too close to the boundaries. This exclusion is controlled by the _bw_std_edge parameter, which defines the distance that a peak must be from the edge in order to be retained (in units of standard deviation; set to 1.0 by default). Therefore, although the algorithm is functioning as intended, users should be careful when interpreting aperiodic parameters in datasets where lowfrequency oscillatory activity might be expected.

In line with the reviewer’s suggestion we have added a version of specparam to the paper.

We thank reviewer 2 for pointing out two studies that used a time-resolved approach to spectral parameterisation. We have updated the text accordingly:

Although a similar approach has been used to track temporal dynamics in sleep and resting state (e.g., Wilson et al., 2022; Ameen et al., 2024), as well as in task-based contexts (e.g., Barrie et al., 1996; Preston et al., 2025), its specific application to working memory paradigms remains underexplored.

**Reviewer 3:**

Reviewer 3 notes that the revised manuscript feels less intriguing than the original version. While we understand this concern, we believe this difference arises from a misalignment in expectations regarding the scope and purpose of our study. We think the reviewer is interpreting our work as focusing on whether theta activity is elicited in a paradigm that reliably produces theta oscillations. In contrast, our study is framed around a working memory task in which, based on prior literature, we expected to observe theta activity but instead found an absence of theta spectral peaks in almost all participants. Note that the absence of theta is already noteworthy in itself, given that theta oscillations are believed to play a crucial role in working memory.

Importantly, Van Engen et al. (2024) have recently reported similar findings:

”While we did not observe load-dependent aperiodic changes over the frontal midline, we did reveal the possibility that previous frontal midline theta results that do not correct for aperiodic activity likely do not reflect theta oscillations. /…/ While our results do not invalidate previous research into extracranial theta oscillations in relation to WM, they challenge popular and widely held beliefs regarding the mechanistic role for theta oscillations to group or segregate channels of information”.

From this perspective, we maintain that the following statements are still justified:

“substantial portion of the changes often attributed to theta oscillations in working memory tasks may be influenced by shifts in the spectral slope of aperiodic activity”

"Note that although no prominent oscillatory peak in the theta range was observed at the group level, and some of this activity could potentially fall within the delta range, similar lowfrequency patterns have often been referred to as 'theta' in previous work, even in the absence of a clear spectral peak"

These formulations are intended to emphasize existing interpretations of changes in low-frequency power as theta oscillations in related research.

Next, Reviewer 3 pointed out that “spectral reflection (peak?) in spectral power plot does not imply that an event is repeating (i..e. oscillatory).” We agree with the reviewer that not every spectral peak implies a true oscillation. To address this, we complemented the power analyses with a measure of rhythmicity (phase autocorrelation function, pACF) after the first round of reviews, and the pACF results were largely similar to those for periodic activity. These results suggest that, in our case, periodic activity is indeed largely oscillatory.

However, we do agree with the reviewer that the term “oscillatory” is not interchangeable with “periodic”. To address this, we reviewed the paper for all appearances of “oscillations”, “oscillatory” and related terms, and replaced them with “power”, “spectral” or “periodic activity” where appropriate (all changes are marked in red in the latest version of the manuscript).

Examples of corrections:

“Changes in aperiodic activity appear as low-frequency oscillations in baseline-corrected time-frequency plots” → low-frequency power

“The periodic component includes only the parameterised oscillatory peak” → spectral peak

“FOOOF decomposition may miss low-frequency oscillations near the edges of the spectrum” → low-frequency peaks

We disagree with the reviewer’s assertion that the subtitle “Aperiodic parameters are largely independent of oscillatory activity” is misleading for a methods oriented paper. Namely, the full subtitle is “Rhythmicity analysis reveals aperiodic parameters are largely independent of oscillatory activity”. Since rhythmicity is a phase-based measure that requires repeating dynamics and is therefore indicative of oscillations, we believe this phrasing is technically accurate.

Finally, we would like to emphasise our contribution once again. Our analyses of rhythmicity, spectrally parameterised power, and baseline-corrected power offer different perspectives on the data. Each of these analyses may lead to different interpretations, but performing all of them on the same data provides a more comprehensive insight into what is actually going on in the data.

Our findings demonstrate that conclusions drawn from a single analytical approach may be incomplete or misleading. For example, as we discuss in the paper, many studies examine theta-gamma coupling in scalp EEG during n-back tasks without first establishing whether theta activity genuinely oscillates (e.g. Rajji et al., 2016). The absence of true theta oscillations would undermine the validity of such analyses. Our multifaceted approach provides researchers with a systematic framework for validating oscillatory assumptions before proceeding with more complex analyses.